# Explicit Discovery of Nonlinear Symmetries from Dynamic Data

**Lexiang Hu** [1]   **Yikang Li** [1]   **Zhouchen Lin** [1 2 3]

## Abstract

Symmetry is widely applied in problems such as the design of equivariant networks and the discovery of governing equations, but in complex scenarios, it is not known in advance. Most previous symmetry discovery methods are limited to linear symmetries, and recent attempts to discover nonlinear symmetries fail to explicitly get the Lie algebra subspace. In this paper, we propose LieNLSD, which is, to our knowledge, the first method capable of determining the number of infinitesimal generators with nonlinear terms and their explicit expressions. We specify a function library for the infinitesimal group action and aim to solve for its coefficient matrix, proving that its prolongation formula for differential equations, which governs dynamic data, is also linear with respect to the coefficient matrix. By substituting the central differences of the data and the Jacobian matrix of the trained neural network into the infinitesimal criterion, we get a system of linear equations for the coefficient matrix, which can then be solved using SVD. On top quark tagging and a series of dynamic systems, LieNLSD shows qualitative advantages over existing methods and improves the long rollout accuracy of neural PDE solvers by over $20\%$ while applying to guide data augmentation. Code and data are available at https://github.com/hulx2002/LieNLSD.

## 1. Introduction

From traditional mathematical physics to deep learning, symmetry plays an important role. In differential equations, symmetries can assist in integration by reducing the order (Olver, 1993; McLachlan, 1995; Ibragimov, 1999; Hydon,

[1]State Key Lab of General AI, School of Intelligence Science and Technology, Peking University [2]Institute for Artificial Intelligence, Peking University [3]Pazhou Laboratory (Huangpu), Guangzhou, Guangdong, China. Correspondence to: Zhouchen Lin <zlin@pku.edu.cn>.

*Proceedings of the $42^{nd}$ International Conference on Machine Learning*, Vancouver, Canada. PMLR 267, 2025. Copyright 2025 by the author(s).

2000; Bluman & Anco, 2008; Bluman, 2010). Equivariant networks embed symmetries into their structure, achieving better performance on specific tasks (Zaheer et al., 2017; Weiler et al., 2018b;a; Kondor & Trivedi, 2018; Wang et al., 2021; Finzi et al., 2021; Satorras et al., 2021). Furthermore, symmetries can guide the discovery of governing equations from data (Yang et al., 2024b). However, these methods all require prior knowledge of symmetries, which is challenging when dealing with complex tasks and messy datasets. Therefore, automatically discovering symmetries from data has become an important topic.

Most existing symmetry discovery methods can successfully identify linear symmetries in the observation space (Zhou et al., 2021; Dehmamy et al., 2021; Moskalev et al., 2022; Yang et al., 2023). However, they are unable to discover nonlinear symmetries, which are common in dynamical systems. For example, in the case of the wave equation $u_{tt} = u_{xx} + u_{yy}$, these methods can only find Lorentz symmetry and scaling symmetry, while missing translation symmetry and special conformal symmetry.

Recent works have attempted to explore nonlinear symmetries. LaLiGAN (Yang et al., 2024a) uses an encoder-decoder architecture to map the observation space to a latent space, making nonlinear symmetries in the observation space appear as linear symmetries in the latent space. However, since the encoder and decoder are black boxes, they cannot explicitly discover the nonlinear symmetries in the observation space. Ko et al. (2024) and Shaw et al. (2024) propose methods to directly find non-affine symmetries from observed data. Nevertheless, Ko et al. (2024) use an MLP to model the infinitesimal generators, which suffers from the same interpretability issues as LaLiGAN, while Shaw et al. (2024) are limited to one-parameter group symmetries.

In this paper, we propose the Lie algebra based NonLinear Symmetry Discovery method (LieNLSD), which can determine the number of infinitesimal generators and explicitly provide their expressions containing nonlinear terms. Our setting assumes that the dataset originates from a dynamic system governed by differential equations, with common forms including first-order dynamic systems $u_t = f(u')$ and second-order dynamical systems $u_{tt} = f(u')$, where $u'$ represents $u$ and its spatial derivatives. This is a general setting, as static systems governed by algebraic equations are

also included (i.e., zero-order dynamic systems). We specify a function library $\Theta$ for the infinitesimal group action, which may include nonlinear terms. Then the infinitesimal group action can be expressed in the form of $W\Theta$. Our ultimate goal is to solve for the coefficient matrix $W$ of $\Theta$, thereby explicitly obtaining the expression for the infinitesimal group action.

We first prove that the prolonged infinitesimal group action is linear with respect to $\text{vec}(W)$, and thus the infinitesimal criterion for the symmetry group of differential equations is also linear with respect to $\text{vec}(W)$. We then estimate derivatives from the dataset using the central difference method and obtain the Jacobian matrix of differential equations via automatic differentiation from the trained neural network. Substituting these into the infinitesimal criterion, we derive a system of linear equations for $\text{vec}(W)$, whose solution space is found based on Singular Value Decomposition (SVD). Finally, we apply the Linearized Alternating Direction Method with Adaptive Penalty (LADMAP) (Lin et al., 2011) to sparsify the infinitesimal generators for intuitive and interpretable results.

In summary, our contributions are as follows: (1) we propose LieNLSD, a novel pipeline for discovering nonlinear symmetries from dynamic data, which is the first method capable of determining the number of nonlinear infinitesimal generators and explicitly solving for their expressions to our knowledge; (2) we prove that if the infinitesimal group action is linear with respect to the coefficient matrix $W$, then its prolongation formula is also linear with respect to $W$; (3) we construct and solve a system of linear equations for the coefficient matrix $\text{vec}(W)$ of the infinitesimal group action based on the infinitesimal criterion; (4) we apply LADMAP to the sparsification of infinitesimal generators to enhance the intuitiveness and interpretability of the results; (5) the experimental results on top quark tagging and a series of dynamic systems confirm the qualitative advantages of LieNLSD over existing methods; (6) we apply LieNLSD to guide data augmentation for neural PDE solvers, improving their long rollout accuracy by over 20%.

## 2. Preliminary

Before discussing related works and methods, we first briefly introduce some preliminary knowledge of Lie groups and differential equations. For readers who are not familiar with the relevant theory, we strongly recommend referring to Appendix A or the textbook (Olver, 1993) for a detailed version, and Appendix C for concrete examples.

**Group actions and infinitesimal group actions.** A Lie group $G$ is a mathematical object that is both a smooth manifold and a group. The Lie algebra $\mathfrak{g}$ serves as the tangent space at the identity of the Lie group. A Lie algebra

element $\mathbf{v} \in \mathfrak{g}$ can be associated with a Lie group element $g \in G$ through the exponential map $\exp : \mathfrak{g} \to G$, i.e., $g = \exp(\mathbf{v})$. Representing the space of the Lie algebra with a basis $\{\mathbf{v}_1, \mathbf{v}_2, \ldots, \mathbf{v}_r\}$, in the neighborhood of the identity element, we have $g = \exp\left(\sum_{i=1}^{r} \epsilon^i \mathbf{v}_i\right)$, where $\epsilon = (\epsilon^1, \epsilon^2, \ldots, \epsilon^r) \in \mathbb{R}^r$ are the coefficients. The group $G$ acts on a vector space $X = \mathbb{R}^n$ via a group action $\Psi : G \times X \to X$. Correspondingly, **the infinitesimal group action** of $\mathbf{v} \in \mathfrak{g}$ at $x \in X$ is defined as:

$$\psi(\mathbf{v})|_x = \left.\frac{\mathrm{d}}{\mathrm{d}\epsilon}\right|_{\epsilon=0} \Psi(\exp(\epsilon\mathbf{v}), x) \cdot \nabla. \tag{1}$$

We abbreviate $\Psi(g, x)$ and $\psi(\mathbf{v})$ as $g \cdot x$ and $\mathbf{v}$, respectively, when the context is clear. We refer to the infinitesimal group actions of the Lie algebra basis as **infinitesimal generators**.

This is a more general definition that encompasses the nonlinear case. Previous works focusing on linear symmetries (Moskalev et al., 2022; Desai et al., 2022; Yang et al., 2023) commonly use the group representation $\rho_X : G \to \text{GL}(n)$ to characterize group transformations, i.e., $\forall g \in G, x \in X : \Psi(g, x) = \rho_X(g)x$. The corresponding Lie algebra representation $\mathrm{d}\rho_X : \mathfrak{g} \to \mathfrak{gl}(n)$ satisfies $\forall \mathbf{v} \in \mathfrak{g} : \rho_X(\exp(\mathbf{v})) = \exp(\mathrm{d}\rho_X(\mathbf{v}))$. Then, **the relationship between infinitesimal group actions and Lie algebra representations** is:

$$\forall \mathbf{v} \in \mathfrak{g}, x \in X : \quad \psi(\mathbf{v})|_x = \mathrm{d}\rho_X(\mathbf{v})x \cdot \nabla. \tag{2}$$

A concrete example of group actions and infinitesimal group actions is provided in Appendix C.1.

**Symmetries of differential equations.** Before defining the symmetries of differential equations, we first explain how groups act on functions. Let $f : X \to U$ be a function, with its graph defined as $\Gamma_f = \{(x, f(x)) : x \in X\} \subset X \times U$. Suppose a Lie group $G$ acts on $X \times U$. Then, the transform of $\Gamma_f$ by $g \in G$ is given by $g \cdot \Gamma_f = \{(\tilde{x}, \tilde{u}) = g \cdot (x, u) : (x, u) \in \Gamma_f\}$. We call a function $\tilde{f} : X \to U$ **the transform of** $f$ **by** $g$, denoted as $\tilde{f} = g \cdot f$, if its graph satisfies $\Gamma_{\tilde{f}} = g \cdot \Gamma_f$. Note that $\tilde{f}$ may not always exist, but in this paper, we only consider cases where it does. A concrete example is provided in Appendix C.2.

The solution of a differential equation is expressed in the form of a function. Intuitively, the symmetries of a differential equation describe how one solution can be transformed into another. Formally, let $\mathscr{S}$ be a system of differential equations, with the independent and dependent variable spaces denoted by $X$ and $U$, respectively. Suppose that a Lie group $G$ acts on $X \times U$. We call $G$ **the symmetry group of** $\mathscr{S}$ if, for any solution $f : X \to U$ of $\mathscr{S}$, and $\forall g \in G$, the transformed function $g \cdot f : X \to U$ is another solution of $\mathscr{S}$.

*Table 1.* Comparison of LieNLSD and other symmetry discovery methods.

| Applicability | L-conv | LieGG | LieGAN | LaLiGAN | Ko et al. | Shaw et al. | LieNLSD (Ours) |
|---|---|---|---|---|---|---|---|
| Nonlinear? | × | × | × | ✓ | ✓ | ✓ | ✓ |
| Explicit? | ✓ | ✓ | ✓ | × | × | ✓ | ✓ |
| Determine Lie algebra subspace dimension? | × | ✓ | × | × | × | × | ✓ |
| Discover Lie algebra subspace? | ✓ | ✓ | ✓ | ✓ | ✓ | × | ✓ |

**Prolongation.** To further study the symmetries of differential equations, we need to "prolong" the group action on the space of independent and dependent variables to the space of derivatives. Given a function $f : X \to U$, where $X = \mathbb{R}^p$ and $U = \mathbb{R}^q$, it has $q \cdot p_k = q \cdot \binom{p+k-1}{k}$ distinct $k$-th order derivatives $u_J^\alpha = \partial_J f^\alpha(x) = \frac{\partial^k f^\alpha(x)}{\partial x^{j_1} \partial x^{j_2} \dots \partial x^{j_k}}$, where $\alpha \in \{1, \dots, q\}$, $J = (j_1, \dots, j_k)$, and $j_i \in \{1, \dots, p\}$. We denote the space of all $k$-th order derivatives as $U_k = \mathbb{R}^{q \cdot p_k}$ and the space of all derivatives up to order $n$ as $U^{(n)} = U \times U_1 \times \dots U_n = \mathbb{R}^{q \cdot p^{(n)}}$, where $p^{(n)} = \binom{p+n}{n}$. Then, **the $n$-th prolongation of $f$**, $\mathrm{pr}^{(n)} f : X \to U^{(n)}$, is defined as $\mathrm{pr}^{(n)} f(x) = u^{(n)}$, where $u_J^\alpha = \partial_J f^\alpha(x)$.

Based on the above concepts, the $n$-th order system of differential equations $\mathscr{S}$ can be formalized as $F(x, u^{(n)}) = 0$, where $F : X \times U^{(n)} \to \mathbb{R}^l$. A smooth function $f : X \to U$ is a solution of $\mathscr{S}$ if it satisfies $F(x, \mathrm{pr}^{(n)} f(x)) = 0$.

Below, we explain how to prolong the group action on $X \times U$ to $X \times U^{(n)}$. Let $G$ be a Lie group acting on the space of independent and dependent variables $X \times U$. Given a point $(x_0, u_0^{(n)}) \in X \times U^{(n)}$, suppose that a smooth function $f : X \to U$ satisfies $u_0^{(n)} = \mathrm{pr}^{(n)} f(x_0)$. Then, **the $n$-th prolongation of $g \in G$ at the point** $(x_0, u_0^{(n)})$ is defined as $\mathrm{pr}^{(n)} g \cdot (x_0, u_0^{(n)}) = (\tilde{x}_0, \tilde{u}_0^{(n)})$, where $(\tilde{x}_0, \tilde{u}_0) = g \cdot (x_0, u_0)$ and $\tilde{u}_0^{(n)} = \mathrm{pr}^{(n)} (g \cdot f)(\tilde{x}_0)$.

Note that by the chain rule, the definition of $\mathrm{pr}^{(n)} g$ depends only on $(x_0, u_0^{(n)})$ and is independent of the choice of $f$. Similarly, **the $n$-th prolongation of $\mathbf{v} \in \mathfrak{g}$ at the point** $(x, u^{(n)}) \in X \times U^{(n)}$ is defined as:

$$\mathrm{pr}^{(n)} \mathbf{v} \Big|_{(x, u^{(n)})} = \frac{\mathrm{d}}{\mathrm{d}\epsilon} \Big|_{\epsilon=0} \left\{ \mathrm{pr}^{(n)} [\exp(\epsilon \mathbf{v})] \cdot (x, u^{(n)}) \right\} \cdot \nabla. \tag{3}$$

Concrete examples of prolongation are provided in Appendix C.3.

## 3. Related Work

**Symmetry discovery.** Some previous works have attempted to discover symmetries from data. Early methods parameterize group symmetries as part of network training (Benton et al., 2020; Zhou et al., 2021; Romero & Lohit, 2022; van der Ouderaa et al., 2024). However, they have

strong restrictions on the type of group symmetries—they require prior knowledge of the parameterized form of group actions or are limited to subgroups of a given group.

Subsequent works use Lie group and Lie algebra representations to characterize group symmetries. L-conv (Dehmamy et al., 2021) proposes the Lie algebra convolutional network, which serves as a building block for constructing group-equivariant architectures and learns the Lie algebra basis. LieGG (Moskalev et al., 2022) constructs a polarization matrix based on the trained network and training data, extracting the Lie algebra basis from it. SymmetryGAN (Desai et al., 2022) and LieGAN (Yang et al., 2023) use generative adversarial training to align data distributions before and after transformations, where the generator generates group transformations from the Lie algebra basis. As shown in Equation (2), Lie group and Lie algebra representations can only describe linear group actions, which limits these methods to discovering linear symmetries.

Recently, some works have explored the discovery of nonlinear group symmetries. LaLiGAN (Yang et al., 2024a) learns a mapping from the observation space to a latent space and extends the LieGAN approach to discover linear group symmetries in the latent space. Ko et al. (2024) model the infinitesimal generators using an MLP and optimize the validity score of the data transformed through ODE integration. Shaw et al. (2024) propose an efficient method for detecting non-affine group symmetries. However, LaLiGAN and Ko et al. (2024) fail to explicitly provide the infinitesimal group actions on the observation space. Shaw et al. (2024) are limited to discovering a single Lie algebra element rather than the subspace it belongs to. Furthermore, these methods cannot accurately determine the number of infinitesimal generators, i.e., the dimension of the Lie algebra subspace.

We compare our LieNLSD with other symmetry discovery methods in Table 1. To our knowledge, LieNLSD is the first method capable of accurately determining the number of nonlinear infinitesimal generators and explicitly solving for their expressions.

Note that from an implementation perspective, the types of PDE symmetries discovered by LieNLSD (ours) and LaLiGAN (Yang et al., 2024a) differ. Taking $X = \mathbb{R}^2, U = \mathbb{R}$ as an example (e.g. $u(x, y)$ represents a planar image), the symmetries found by LieNLSD act pointwise on $X \times$

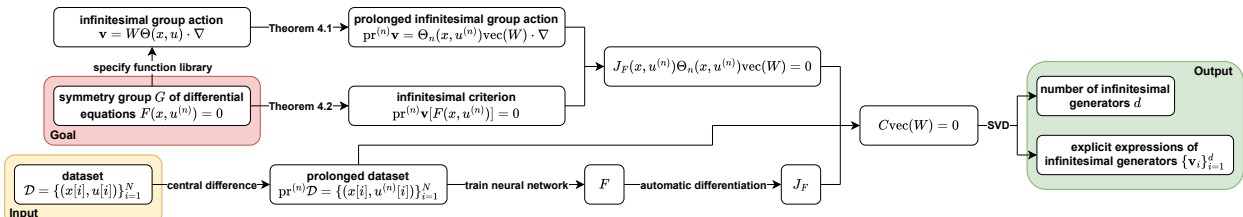

*Figure 1.* Pipeline of LieNLSD.

$U = \mathbb{R}^3$. Such symmetries are commonly referred to in the literature as "Lie point symmetries". On the other hand, the symmetries found by LaLiGAN are defined over the entire discretized field. Specifically, if $u$ is a field on a $100 \times 100$ grid, then LaLiGAN's symmetries act on $\mathbb{R}^{100 \times 100}$ (see the reaction-diffusion dataset in the original paper of LaLiGAN for details). For PDEs, the setting of Lie point symmetries (which act pointwise on both coordinates and the field) is more common, as the search space is significantly reduced compared to symmetries defined over the entire discretized field, and the physical interpretation is more intuitive.

**Applications of symmetry.** Symmetry is widely applied in problems such as the design of equivariant networks and the discovery of governing equations. We summarize related works in Appendix D.

## 4. Method

We formalize the problem as follows: given dynamic data governed by differential equations, we aim to determine the number of infinitesimal generators and their explicit expressions. In Section 4.1, we provide a flexible function library for the infinitesimal group action and present the prolongation formula in Theorem 4.1, which is linear with respect to the coefficient matrix $W$. In Section 4.2, we introduce the infinitesimal criterion for the symmetry group of differential equations in Theorem 4.2. It is combined with central differences of the data and automatic differentiation from the trained neural network to construct a system of linear equations for $\mathrm{vec}(W)$, whose solution space corresponds to the infinitesimal generators. In Section 4.3, we summarize the overall algorithm. In Section 4.4, we sparsify the infinitesimal generators for better intuitiveness and interpretability. We present the pipeline of LieNLSD in Figure 1.

### 4.1. Prolongation Formula of Infinitesimal Group Actions

To explicitly obtain the infinitesimal group action on $X \times U = \mathbb{R}^p \times \mathbb{R}^q$, we define $\mathbf{v} = W\Theta(x, u) \cdot \nabla$. Here, $\Theta(x, u) \in \mathbb{R}^{r \times 1}$ is a predefined function library, and $W = [W_1, W_2, \ldots, W_{p+q}]^\top \in \mathbb{R}^{(p+q) \times r}$ is a coefficient matrix to be determined. For example, in the

case where $p = 1$ and $q = 1$, we can specify the function library to include terms up to the second order, $\Theta(x, u) = [1, x, u, x^2, u^2, xu]^\top$. Clearly, $\mathbf{v}$ is linear with respect to $W$, i.e., $\mathbf{v} = \Theta_0(x, u)\mathrm{vec}(W)$, where $\Theta_0(x, u) = \mathrm{diag}[\Theta(x, u)^\top, \ldots, \Theta(x, u)^\top] \in \mathbb{R}^{(p+q) \times ((p+q) \cdot r)}$. To study the symmetries of differential equations, we are interested in whether $\mathrm{pr}^{(n)}\mathbf{v}$ remains linear with respect to $W$. In fact, the following theorem will provide the answer.

**Theorem 4.1.** *Let $G$ be a Lie group acting on $X \times U = \mathbb{R}^p \times \mathbb{R}^q$, with its corresponding Lie algebra $\mathfrak{g}$. Assume that the infinitesimal group action of $\mathbf{v} \in \mathfrak{g}$ takes the following form:*

$$\mathbf{v} = W\Theta(x, u) \cdot \nabla \tag{4}$$

$$= \begin{bmatrix} W_1 & W_2 & \cdots & W_{p+q} \end{bmatrix}^\top \Theta(x, u) \cdot \nabla \tag{5}$$

$$= \sum_{i=1}^p \Theta(x, u)^\top W_i \frac{\partial}{\partial x^i} + \sum_{\alpha=1}^q \Theta(x, u)^\top W_{p+\alpha} \frac{\partial}{\partial u^\alpha}, \tag{6}$$

*where $W \in \mathbb{R}^{(p+q) \times r}$ and $\Theta(x, u) \in \mathbb{R}^{r \times 1}$. Then, the $n$-th prolongation of $\mathbf{v}$ is:*

$$\mathrm{pr}^{(n)}\mathbf{v} = \mathbf{v} + \sum_{\alpha=1}^q \sum_J \phi_\alpha^J(x, u^{(n)}) \frac{\partial}{\partial u_J^\alpha}, \tag{7}$$

*where $J = (j_1, \ldots, j_k)$, with $j_i \in \{1, \ldots, p\}$ and $k \in \{1, \ldots, n\}$. The coefficients $\phi_\alpha^J$ are given by:*

$$\phi_\alpha^J(x, u^{(n)}) = -\sum_{i=1}^p \sum_{I \subset J} u_{I,i}^\alpha D_{J \setminus I} \Theta^\top W_i + D_J \Theta^\top W_{p+\alpha}, \tag{8}$$

*where $u_{J,i}^\alpha = \frac{\partial u_J^\alpha}{\partial x^i} = \frac{\partial^{k+1} u^\alpha}{\partial x^i \partial x^{j_1} \ldots \partial x^{j_k}}$, $J \setminus I$ denotes the set difference, and $D_J$ represents the total derivative.*

The proof of Theorem 4.1 is provided in Appendix B.2. Note the distinction between total derivatives and partial derivatives. For a smooth function $P(x, u^{(n)})$, they are related by $D_i P = \frac{\partial P}{\partial x^i} + \sum_{\alpha=1}^q \sum_J u_{J,i}^\alpha \frac{\partial P}{\partial u_J^\alpha}$. Equations (7) and (8) indicate that $\mathrm{pr}^{(n)}\mathbf{v}$ remains linear with respect to $W$, thus it can be expressed as:

$$\mathrm{pr}^{(n)}\mathbf{v} = \Theta_n(x, u^{(n)})\mathrm{vec}(W) \cdot \nabla, \tag{9}$$

where $\Theta_n(x, u^{(n)}) \in \mathbb{R}^{(p+q\cdot p^{(n)})\times((p+q)\cdot r)}$. We provide a concrete example of the construction of $\Theta_n$ in Appendix C.4.

---

**Algorithm 1** LieNLSD
---
**Input:** Dataset $\mathcal{D} = \{(x[i], u[i])\}_{i=1}^N$, prolongation order $n$, function library $\Theta : X \times U \to \mathbb{R}^{r\times 1}$, sample size $M$, thresholds $\epsilon_1, \epsilon_2$.
**Output:** Number of infinitesimal generators $d$, explicit expressions of infinitesimal generators $\{\mathbf{v}_i\}_{i=1}^d$.
**Execute:**
Estimate the derivatives of $u$ with respect to $x$ using the central difference method, resulting in the prolonged dataset $\mathrm{pr}^{(n)}\mathcal{D} = \{(x[i], u^{(n)}[i])\}_{i=1}^N$.
Separate the variables of $\mathrm{pr}^{(n)}\mathcal{D} = S_{in} \cup S_{out}$ into input and output features (for example, for a first-order dynamic system, $S_{in} = \{u'\}$ and $S_{out} = \{u_t\}$), and train a neural network to fit the mapping $f : S_{in} \to S_{out}$.
Construct $\Theta_n : X \times U^{(n)} \to \mathbb{R}^{(p+q\cdot p^{(n)})\times((p+q)\cdot r)}$ from $\Theta$ by Equations (7) to (9).
Apply automatic differentiation to the trained neural network, obtaining $J_F : X \times U^{(n)} \to \mathbb{R}^{l\times(p+q\cdot p^{(n)})}$.
**if** $\frac{M\cdot l}{(p+q)\cdot r} < \epsilon_1$ **then**
    Compute $C \in \mathbb{R}^{(M\cdot l)\times((p+q)\cdot r)}$ by Equation (13).
    Perform SVD on $C$, the number of singular values less than $\epsilon_2$ is $d$, and the corresponding right singular vector matrix is $Q \in \mathbb{R}^{((p+q)\cdot r)\times d}$.
**else**
    Initialize $C^\top C \leftarrow \mathbf{0}_{((p+q)\cdot r)\times((p+q)\cdot r)}$.
    **for** $i = 1$ **to** $M$ **do**
        $C_i \leftarrow (J_F\Theta_n)(x[i], u^{(n)}[i])$.
        $C^\top C \leftarrow C^\top C + C_i^\top C_i$.
    **end for**
    Perform SVD on $C^\top C$, the number of singular values less than $\epsilon_2^2$ is $d$, and the corresponding right singular vector matrix is $Q \in \mathbb{R}^{((p+q)\cdot r)\times d}$.
**end if**
**for** $i = 1$ **to** $d$ **do**
    $W_i \leftarrow Q[:, i].reshape(p + q, r)$.
    $\mathbf{v}_i \leftarrow W_i\Theta(x, u) \cdot \nabla$.
**end for**
**Return** $d, \{\mathbf{v}_i\}_{i=1}^d$.

---

## 4.2. Explicit Discovery of Infinitesimal Group Actions

Our goal is to determine the coefficient matrix $W$ of the infinitesimal group action $\mathbf{v} = W\Theta(x, u) \cdot \nabla$, and we have already derived the formula for its prolongation $\mathrm{pr}^{(n)}\mathbf{v}$. Theorem 2.31 in the textbook (Olver, 1993) relates the symmetry group of a differential equation to its prolonged infinitesimal group action, which is restated as follows.

**Theorem 4.2.** *Suppose $\mathscr{S}$:*

$$F_\nu(x, u^{(n)}) = 0, \quad \nu = 1, \dots, l, \qquad (10)$$

*is a system of differential equations, where $F : X \times U^{(n)} \to \mathbb{R}^l$ is of full rank (the Jacobian matrix $J_F(x, u^{(n)}) = \left(\frac{\partial F_\nu}{\partial x^i}, \frac{\partial F_\nu}{\partial u_J^\alpha}\right) \in \mathbb{R}^{l\times(p+q\cdot p^{(n)})}$ is of rank $l$ whenever $F(x, u^{(n)}) = 0$). A Lie group $G$ acts on $X \times U$, with its corresponding Lie algebra $\mathfrak{g}$. If $\forall \mathbf{v} \in \mathfrak{g}$:*

$$\mathrm{pr}^{(n)}\mathbf{v}[F_\nu(x, u^{(n)})] = 0, \quad \nu = 1, \dots, l, \qquad (11)$$

*whenever $F(x, u^{(n)}) = 0$, then $G$ is a symmetry group of $\mathscr{S}$.*

Substituting Equation (9) into Equation (11), we find that the infinitesimal criterion is also linear with respect to $W$:

$$J_F(x, u^{(n)})\Theta_n(x, u^{(n)})\mathrm{vec}(W) = 0, \qquad (12)$$

whenever $F(x, u^{(n)}) = 0$. As shown in Equations (7) to (9), $\Theta_n$ can be computed from $\Theta$, which is manually specified. Therefore, as long as $J_F$ and the data samples of $(x, u^{(n)})$ are available, $W$ can be solved for.

In practice, we use the central difference method to estimate arbitrary-order derivatives of $u$ with respect to $x$ from the dataset $\mathcal{D} = \{(x[i], u[i])\}_{i=1}^N$, obtaining the prolonged dataset $\mathrm{pr}^{(n)}\mathcal{D} = \{(x[i], u^{(n)}[i])\}_{i=1}^N$. We then separate the variables in $\mathrm{pr}^{(n)}\mathcal{D}$ into input and output features, fitting the mapping of this supervised learning problem using a neural network. For example, a first-order dynamic system is governed by a PDE of the form $u_t = f(u')$ (Kantamneni et al., 2024), where $u'$ represents the set of $u$ and its spatial derivatives. In this case, the differential equation can be expressed as $F(x, u^{(n)}) = f(u') - u_t = 0$, and a neural network is used to fit $f$. The Jacobian matrix $J_F$ can then be obtained by applying automatic differentiation to the trained neural network.

Equation (12) holds for each data sample in $\mathrm{pr}^{(n)}\mathcal{D}$. We sample $M$ points from it for symmetry discovery, resulting in the following system of linear equations:

$$C\mathrm{vec}(W) = \begin{bmatrix} (J_F\Theta_n)(x[1], u^{(n)}[1]) \\ (J_F\Theta_n)(x[2], u^{(n)}[2]) \\ \vdots \\ (J_F\Theta_n)(x[M], u^{(n)}[M]) \end{bmatrix} \mathrm{vec}(W) = 0.$$

$$(13)$$

Perform Singular Value Decomposition (SVD) on $C$: $C\mathrm{vec}(W) = U \begin{bmatrix} \Sigma & 0 \\ 0 & 0 \end{bmatrix} \begin{bmatrix} P^\top \\ Q^\top \end{bmatrix} \mathrm{vec}(W) = 0$. Then, the null space of $C$ is the solution space of $\mathrm{vec}(W)$, with the general solution being $\mathrm{vec}(W) = Q\beta$. In other words, the column vectors of $Q \in \mathbb{R}^{((p+q)\cdot r)\times d}$ form a basis for the space of $\mathrm{vec}(W)$, and $\beta \in \mathbb{R}^{d\times 1}$ is the coordinate vector. Overall,

the number of zero singular values of $C$ represents the number of infinitesimal generators, and the corresponding right singular vectors are their coefficient matrices. Therefore, compared with recent methods for discovering nonlinear symmetries (Yang et al., 2024a; Ko et al., 2024; Shaw et al., 2024), our approach can mathematically determine the dimension of the Lie algebra subspace and explicitly reveal the infinitesimal generators.

When the hyperparameter $M \gg r$, for efficiency, we compute $C^\top C \in \mathbb{R}^{((p+q) \cdot r) \times ((p+q) \cdot r)}$ instead of $C \in \mathbb{R}^{(M \cdot l) \times ((p+q) \cdot r)}$. Specifically, $C^\top C = \sum_{i=1}^{M} C_i^\top C_i$, where $C_i = (J_F \Theta_n)(x[i], u^{(n)}[i])$. Note that if $C = U \Sigma V^\top$, then $C^\top C = V \Sigma^2 V^\top$. Therefore, performing SVD on $C^\top C$ will give us the singular values and right singular vectors of $C$ that we are interested in.

### 4.3. The Overall Algorithm

We now summarize the LieNLSD method in Algorithm 1. LieNLSD consists of two main stages: neural network training and symmetry discovery, which are decoupled from each other. In other words, the symmetry discovery procedure is plug-and-play for a trained neural network. The time and space complexity analysis of LieNLSD is provided in Appendix E. Although LieNLSD starts from the setting of nonlinear symmetries and dynamic data, it can still handle cases of linear/affine symmetries and static data (governed by the arithmetic equation $F(x) = 0$), which are discussed in Appendix F.

### 4.4. Basis Sparsification

In Algorithm 1, SVD guarantees the orthogonality of the basis, but it does not ensure its sparsity. For example, if the ground truth basis vectors are $\frac{\sqrt{2}}{2}[1, 0, 1, 0]^\top$ and $\frac{\sqrt{2}}{2}[0, -1, 0, 1]^\top$, we might solve for $\frac{1}{2}[1, -1, 1, 1]^\top$ and $\frac{1}{2}[1, 1, 1, -1]^\top$. Although they span the same subspace and are both orthogonal, the latter lacks intuitiveness and interpretability. Therefore, we aim to find an orthogonal transformation that makes the transformed basis as sparse as possible, which can be formalized as:

$$\min_{R} \|QR\|_{1,1}, \quad \text{s.t.} \quad R^\top R = I, \qquad (14)$$

where $Q \in \mathbb{R}^{((p+q) \cdot r) \times d}$, $R \in \mathbb{R}^{d \times d}$, and the $(1,1)$-norm of a matrix $\|A\|_{1,1} = \sum_{i,j} |A_{ij}|$ is the sum of the absolute values of all its elements. Note that due to the non-smoothness of the objective function, gradient-based optimization methods such as SGD (Robbins & Monro, 1951), Adam (Kingma, 2014), etc., cannot work. We use the Linearized Alternating Direction Method with Adaptive Penalty (LADMAP) (Lin et al., 2011) to solve this constrained optimization problem. By introducing an auxiliary variable $Z \in \mathbb{R}^{((p+q) \cdot r) \times d}$, the

original problem is transformed into:

$$\min_{R, Z} \|Z\|_{1,1}, \quad \text{s.t.} \quad R^\top R = I, \quad Z = QR. \qquad (15)$$

As shown in Algorithm 2, during the iterative process, we alternately update $R$ and $Z$. Here, the 2-norm of a matrix $\|A\|_2 = \sigma_{max}(A)$ is the largest singular value, and the $\infty$-norm of a matrix $\|A\|_\infty = \max_i \sum_j |A_{ij}|$ is the maximum row sum. The detailed derivation is provided in Appendix G.

---

**Algorithm 2** Basis sparsification

---

**Input:** Basis matrix $Q \in \mathbb{R}^{((p+q) \cdot r) \times d}$.
**Output:** Sparse basis matrix $Q^* \in \mathbb{R}^{((p+q) \cdot r) \times d}$.
Initialize $\epsilon_1 > 0$, $\epsilon_2 > 0$, $\beta_{max} \gg \beta_0 > 0$, $\rho_0 \geq 1$, $\eta_R > \|Q\|_2^2$, $\eta_Z > 1$, $R_0 = I_d$, $Z_0 = QR_0$, $\Lambda_0 = \mathbf{0}_{((p+q) \cdot r) \times d}$, and $k \leftarrow 0$.
**repeat**
  Update $R_{k+1} = UV^\top$, where $U\Sigma V^\top$ is the SVD of $R_k - \frac{Q^\top (\Lambda_k + \beta_k (QR_k - Z_k))}{\beta_k \eta_R}$.
  Update $Z_{k+1} = \mathcal{S}_{(\beta_k \eta_Z)^{-1}} \left( Z_k + \frac{\Lambda_k + \beta_k (QR_{k+1} - Z_k)}{\beta_k \eta_Z} \right)$, where $\mathcal{S}_\epsilon(x) = \text{sgn}(x) \max(|x| - \epsilon, 0)$ is the soft thresholding operator.
  Update $\Lambda_{k+1} = \Lambda_k + \beta_k (QR_{k+1} - Z_{k+1})$.
  **if** $\beta_k \max(\sqrt{\eta_R} \|R_{k+1} - R_k\|_\infty, \sqrt{\eta_Z} \|Z_{k+1} - Z_k\|_\infty) < \epsilon_2$ **then**
    Set $\rho = \rho_0$.
  **else**
    Set $\rho = 1$.
  **end if**
  Update $\beta_{k+1} = \min(\beta_{max}, \rho \beta_k)$.
  $k \leftarrow k + 1$.
**until** $\|QR_k - Z_k\|_\infty < \epsilon_1$ and $\beta_{k-1} \max(\sqrt{\eta_R} \|R_k - R_{k-1}\|_\infty, \sqrt{\eta_Z} \|Z_k - Z_{k-1}\|_\infty) \leq \epsilon_2$
**Return** $QR_k$.

---

## 5. Experiment

In this section, we evaluate LieNLSD on top quark tagging (Section 5.2) and a series of dynamic systems (Section 5.3). The selection of the quantitative metric and baseline is discussed in Section 5.1. We demonstrate the application of LieNLSD in guiding data augmentation for neural PDE solvers in Section 5.4. The additional experiments are provided in Appendix I.

### 5.1. Quantitative Metric: Grassmann Distance

LieNLSD uses the subspace where the Lie algebra resides to represent symmetries. Therefore, we choose the Grassmann distance, which measures the difference between the computed subspace and the ground truth subspace, as a quantitative metric. Let $Q_1, Q_2 \in \mathbb{R}^{n \times d}$ be the orthogonal bases of two $d$-dimensional subspaces in $\mathbb{R}^n$. Perform SVD

on $Q_1^\top Q_2$ to obtain the singular values $\{\sigma_i\}_{i=1}^d$. The Grassmann distance between the two subspaces is then defined as $d_G(Q_1, Q_2) = \sqrt{\sum_{i=1}^d \theta_i^2}$, where $\theta_i = \arccos \sigma_i$ are the principal angles.

As shown in Table 1, other methods for discovering nonlinear symmetries (Yang et al., 2024a; Ko et al., 2024; Shaw et al., 2024) cannot explicitly provide the subspace where the Lie algebra resides, rendering this quantitative metric inapplicable. We extract linear infinitesimal generators from LieNLSD for comparison with LieGAN (Yang et al., 2023), the state-of-the-art method for explicitly discovering Lie algebra basis. Before calculating the Grassmann distance, we perform QR decomposition on all basis matrices to ensure orthogonality and normalization. Furthermore, we will compare the number of parameters in LieNLSD and LieGAN. The parameter overhead of LieNLSD is mainly focused on the neural network used to fit the mapping, while that of LieGAN is mainly concentrated in the discriminator.

We quantitatively compare LieNLSD with LieGAN on all experiments in this section in Table 2. The randomness of LieNLSD mainly comes from the selection of sample points for symmetry discovery, while that of LieGAN mainly arises from the random seed setting.

*Table 2.* Quantitative comparison of LieNLSD and LieGAN on all experiments in this section. The Grassmann distance is presented in the format of mean $\pm$ std over three runs.

| Dataset | Model | Grassmann distance ($\downarrow$) | Parameters |
|---|---|---|---|
| Top quark tagging | LieNLSD | $(\mathbf{9.20 \pm 1.83}) \times \mathbf{10^{-2}}$ | 97K |
| | LieGAN | $(2.51 \pm 0.41) \times 10^{-1}$ | 321K |
| Burgers' equation | LieNLSD | $(\mathbf{1.26 \pm 0.20}) \times \mathbf{10^{-2}}$ | 81K |
| | LieGAN | $1.58 \pm 0.05$ | 265K |
| Wave equation | LieNLSD | $(\mathbf{1.40 \pm 0.01}) \times \mathbf{10^{-2}}$ | 82K |
| | LieGAN | $2.36 \pm 0.15$ | 266K |
| Schrödinger equation | LieNLSD | $(\mathbf{8.62 \pm 1.31}) \times \mathbf{10^{-2}}$ | 83K |
| | LieGAN | $2.22 \pm 0.05$ | 266K |

### 5.2. Linear Symmetry Discovery

**Top quark tagging.** We first evaluate the ability of LieNLSD to discover linear symmetries on top quark tagging (Kasieczka et al., 2019). The task is to classify hadronic tops from QCD backgrounds. Its input consists of the four-momenta $p_i^\mu = (p_i^0, p_i^1, p_i^2, p_i^3) \in \mathbb{R}^4$ of several jet constituents produced in an event, and the output is the event label (1 for top quark decay, 0 for other events).

This is a static system, so we set the prolongation order to $n = 0$. To discover linear symmetries, the function library is specified as $\Theta(p^\mu) = [p^0, p^1, p^2, p^3]^\top \in \mathbb{R}^{4 \times 1}$. We use an MLP with 3 hidden layers and hidden dimension 200 to fit the mapping. The sample size for symmetry discovery

is $M = 100$. For LieGAN, we set the dimension of the Lie algebra basis to 7, using an MLP with 2 hidden layers and hidden dimension 512 as the discriminator, which is the same setting as in the original paper (Yang et al., 2023). More implementation details are provided in Appendix H.1.

We present the visualization result of LieNLSD on top quark tagging in Figure 2 (in Appendix H.1). LieNLSD obtains 7 nearly zero singular values, which indicates that the number of infinitesimal generators is 7. The corresponding explicit expressions are shown in Table 3. These constitute Lorentz symmetry and scaling symmetry, where $\mathbf{v}_1, \mathbf{v}_3, \mathbf{v}_5$ represent spatial rotations, $\mathbf{v}_2, \mathbf{v}_4, \mathbf{v}_6$ represent Lorentz boosts, and $\mathbf{v}_7$ represents scaling transformations. As shown in Table 2, even for linear symmetry discovery, LieNLSD outperforms LieGAN with fewer parameters. Additionally, LieNLSD can automatically determine the dimension of the Lie algebra subspace, whereas LieGAN requires manual specification.

### 5.3. Nonlinear Symmetry Discovery

We next evaluate the ability of LieNLSD to capture nonlinear symmetries on dynamic data governed by Burgers' equation, the wave equation, and the Schrödinger equation. We generate several initial conditions by randomly sampling the coefficients of the Fourier series. Then, we estimate the spatial derivatives at each point using central differences and numerically integrate the trajectories corresponding to these initial conditions using the fourth-order Runge-Kutta method (RK4). By selecting time and space steps for sampling, we obtain a discrete dataset.

For LieNLSD, the function library is specified as up to second-order terms. We set the prolongation order to $n = 2$, and use the central difference method to estimate all derivatives up to the second order. We train an MLP with 3 hidden layers and hidden dimension 200 to fit the mapping $u_t = f(u^{(2)})$ for first-order dynamic systems, or $u_{tt} = f(u^{(2)})$ for second-order dynamic systems. The sample size for symmetry discovery is $M = 100$. More implementation details of dataset generation and symmetry discovery are provided in Appendix H.2.

**Burgers' equation.** Burgers' equation describes the convection-diffusion phenomenon, which is widely applied in areas such as fluid mechanics, nonlinear acoustics, gas dynamics, and traffic flow. Its potential form is $u_t = u_{xx} + u_x^2$.

We present the visualization result of LieNLSD on Burgers' equation in Figure 3 (in Appendix H.2). LieNLSD obtains 6 nearly zero singular values, which indicates that the number of infinitesimal generators is 6. The corresponding explicit expressions are shown in Table 3. The group actions they generate for the symmetry group are $g_1 \cdot (t, x, u) =$

*Table 3.* Infinitesimal generators found on all experiments in this section by LieNLSD.

| Dataset | Top quark tagging | Burgers' equation | Wave equation | Schrödinger equation |
|---|---|---|---|---|
| Generators | $\mathbf{v}_1 = -p^3\frac{\partial}{\partial p^2} + p^2\frac{\partial}{\partial p^3}$, 
 $\mathbf{v}_2 = p^1\frac{\partial}{\partial p^0} + p^0\frac{\partial}{\partial p^1}$, 
 $\mathbf{v}_3 = -p^3\frac{\partial}{\partial p^1} + p^1\frac{\partial}{\partial p^3}$, 
 $\mathbf{v}_4 = p^3\frac{\partial}{\partial p^0} + p^0\frac{\partial}{\partial p^3}$, 
 $\mathbf{v}_5 = -p^2\frac{\partial}{\partial p^1} + p^1\frac{\partial}{\partial p^2}$, 
 $\mathbf{v}_6 = p^2\frac{\partial}{\partial p^0} + p^0\frac{\partial}{\partial p^2}$, 
 $\mathbf{v}_7 = p^0\frac{\partial}{\partial p^0} + p^1\frac{\partial}{\partial p^1} + p^2\frac{\partial}{\partial p^2} + p^3\frac{\partial}{\partial p^3}$ | $\mathbf{v}_1 = 4t^2\frac{\partial}{\partial t} + 4tx\frac{\partial}{\partial x} - (2t+x^2)\frac{\partial}{\partial u}$, 
 $\mathbf{v}_2 = 2t\frac{\partial}{\partial t} + x\frac{\partial}{\partial x}$, 
 $\mathbf{v}_3 = 2t\frac{\partial}{\partial x} - x\frac{\partial}{\partial u}$, 
 $\mathbf{v}_4 = \frac{\partial}{\partial u}$, 
 $\mathbf{v}_5 = \frac{\partial}{\partial t}$, 
 $\mathbf{v}_6 = \frac{\partial}{\partial x}$ | $\mathbf{v}_1 = 2tx\frac{\partial}{\partial t} + (t^2+x^2-y^2)\frac{\partial}{\partial x} + 2xy\frac{\partial}{\partial y} - xu\frac{\partial}{\partial u}$, 
 $\mathbf{v}_2 = 2ty\frac{\partial}{\partial t} + 2xy\frac{\partial}{\partial x} + (t^2-x^2+y^2)\frac{\partial}{\partial y} - yu\frac{\partial}{\partial u}$, 
 $\mathbf{v}_3 = (t^2+y^2)\frac{\partial}{\partial u}$, $\mathbf{v}_4 = (t^2+2x^2-y^2)\frac{\partial}{\partial u}$, $\mathbf{v}_5 = xy\frac{\partial}{\partial u}$, 
 $\mathbf{v}_6 = (t^2+x^2+y^2)\frac{\partial}{\partial t} + 2tx\frac{\partial}{\partial x} + 2ty\frac{\partial}{\partial y} - tu\frac{\partial}{\partial u}$, 
 $\mathbf{v}_7 = tx\frac{\partial}{\partial u}$, $\mathbf{v}_8 = x\frac{\partial}{\partial t} + t\frac{\partial}{\partial x}$, $\mathbf{v}_9 = y\frac{\partial}{\partial t} + t\frac{\partial}{\partial y}$, 
 $\mathbf{v}_{10} = t\frac{\partial}{\partial t} + x\frac{\partial}{\partial x} + y\frac{\partial}{\partial y}$, $\mathbf{v}_{11} = ty\frac{\partial}{\partial u}$, $\mathbf{v}_{12} = -y\frac{\partial}{\partial x} + x\frac{\partial}{\partial y}$, 
 $\mathbf{v}_{13} = u\frac{\partial}{\partial u}$, $\mathbf{v}_{14} = y\frac{\partial}{\partial u}$, $\mathbf{v}_{15} = x\frac{\partial}{\partial u}$, $\mathbf{v}_{16} = t\frac{\partial}{\partial u}$, 
 $\mathbf{v}_{17} = \frac{\partial}{\partial u}$, $\mathbf{v}_{18} = \frac{\partial}{\partial t}$, $\mathbf{v}_{19} = \frac{\partial}{\partial x}$, $\mathbf{v}_{20} = \frac{\partial}{\partial y}$ | $\mathbf{v}_1 = -2t\frac{\partial}{\partial t} - x\frac{\partial}{\partial x} - y\frac{\partial}{\partial y} + u\frac{\partial}{\partial u} + v\frac{\partial}{\partial v}$, 
 $\mathbf{v}_2 = -v\frac{\partial}{\partial u} + u\frac{\partial}{\partial v}$, 
 $\mathbf{v}_3 = -y\frac{\partial}{\partial x} + x\frac{\partial}{\partial y}$, 
 $\mathbf{v}_4 = \frac{\partial}{\partial t}$, 
 $\mathbf{v}_5 = \frac{\partial}{\partial x}$, 
 $\mathbf{v}_6 = \frac{\partial}{\partial y}$ |

$\left(\frac{t}{1-4\epsilon t}, \frac{x}{1-4\epsilon t}, u + \frac{1}{2}\ln(1-4\epsilon t) - \frac{\epsilon x^2}{1-4\epsilon t}\right)$, $g_2 \cdot (t,x,u) = (e^{2\epsilon}t, e^\epsilon x, u)$, $g_3 \cdot (t,x,u) = (t, x+2\epsilon t, u - \epsilon x - \epsilon^2 t)$, $g_4 \cdot (t,x,u) = (t,x,u+\epsilon)$, $g_5 \cdot (t,x,u) = (t+\epsilon, x, u)$, and $g_6 \cdot (t,x,u) = (t, x+\epsilon, u)$, where $g_i = \exp(\epsilon \mathbf{v}_i)$. The practical meaning is that, if $u = f(t,x)$ is a solution to Burgers' equation, then $u_1 = f\left(\frac{t}{1+4\epsilon t}, \frac{x}{1+4\epsilon t}\right) - \frac{1}{2}\ln(1+4\epsilon t) - \frac{\epsilon x^2}{1+4\epsilon t}$, $u_2 = f(e^{-2\epsilon}t, e^{-\epsilon}x)$, $u_3 = f(t, x-2\epsilon t) - \epsilon x + \epsilon^2 t$, $u_4 = f(t,x) + \epsilon$, $u_5 = f(t-\epsilon, x)$, and $u_6 = f(t, x-\epsilon)$ are also solutions.

**Wave equation.** The wave equation describes how waves propagate through various mediums. Its form in two-dimensional space is $u_{tt} = u_{xx} + u_{yy}$, where $u(t,x,y)$ is the displacement or, more generally, the conserved quantity at time $t$ and position $(x,y)$.

We present the visualization result of LieNLSD on the wave equation in Figure 4 (in Appendix H.2). LieNLSD obtains 20 nearly zero singular values, which indicates that the number of infinitesimal generators is 20. The corresponding explicit expressions are shown in Table 3. There are 10 infinitesimal generators that constitute the conformal symmetry, where $\mathbf{v}_8, \mathbf{v}_9, \mathbf{v}_{12}$ represent Lorentz transformations, $\mathbf{v}_{18}, \mathbf{v}_{19}, \mathbf{v}_{20}$ represent translations, $\mathbf{v}_{10}$ represents scaling transformations, and $\mathbf{v}_1, \mathbf{v}_2, \mathbf{v}_6$ represent special conformal transformations. Additionally, $\mathbf{v}_{13}$ generates the group action $\exp(\epsilon\mathbf{v}_{13})(t,x,y,u) = (t,x,y,e^\epsilon u)$, and $\mathbf{v}_3, \mathbf{v}_4, \mathbf{v}_5, \mathbf{v}_7, \mathbf{v}_{11}, \mathbf{v}_{14}, \mathbf{v}_{15}, \mathbf{v}_{16}, \mathbf{v}_{17}$ can be unified as $\mathbf{v}_\alpha = \alpha(t,x,y)\frac{\partial}{\partial u}$, where $\alpha$ is a solution to the wave equation $\alpha_{tt} = \alpha_{xx} + \alpha_{yy}$. This implies that if $u = f(t,x,y)$ is a solution to the wave equation, then $u_{13} = e^\epsilon f(t,x,y)$ and $u_\alpha = f(t,x,y) + \epsilon\alpha(t,x,y)$ are also solutions.

**Schrödinger equation.** The Schrödinger equation describes the evolution of the quantum state of microscopic particles over time. Its parameterization on a plane in terms of real ($u$) and imaginary ($v$) components is expressed as:

$$\begin{cases} u_t = -0.5(v_{xx} + v_{yy}) + vu^2 + v^3, \\ v_t = 0.5(u_{xx} + u_{yy}) - uv^2 - u^3. \end{cases} \quad (16)$$

We present the visualization result of LieNLSD on the Schrödinger equation in Figure 5 (in Appendix H.2). LieNLSD obtains 6 nearly zero singular values, which indicates that the number of infinitesimal generators is 6. The corresponding explicit expressions are shown in Table 3. Then, if $\mathbf{u} = \mathbf{f}(t, \mathbf{x})$ is a solution to the Schrödinger equation, we can obtain several derived solutions through these infinitesimal generators. Specifically, $\mathbf{v}_1$ represents scaling $\mathbf{u}_1 = e^\epsilon \mathbf{f}(e^{2\epsilon}t, e^\epsilon\mathbf{x})$, $\mathbf{v}_2$ represents rotation in the complex plane $\mathbf{u}_2 = R\mathbf{f}(t, \mathbf{x})$, $\mathbf{v}_3$ represents space rotation $\mathbf{u}_3 = \mathbf{f}(t, R^{-1}\mathbf{x})$, $\mathbf{v}_4$ represents time translation $\mathbf{u}_4 = \mathbf{f}(t-\epsilon, \mathbf{x})$, and $\mathbf{v}_5, \mathbf{v}_6$ represent space translation $\mathbf{u}_{5,6} = \mathbf{f}(t, \mathbf{x} - \boldsymbol{\epsilon})$, where $R \in \mathrm{SO}(2)$ and $\boldsymbol{\epsilon} \in \mathbb{R}^2$.

LieNLSD has already demonstrated qualitative advantages over LieGAN, such as the discovery of nonlinear symmetries and the determination of Lie algebra subspace dimension. Furthermore, we extract the linear infinitesimal generators found by LieNLSD on the aforementioned three dynamic datasets and quantitatively compare their accuracy with the results from LieGAN. The discriminator of Lie-GAN is set as an MLP with 2 hidden layers and hidden dimension 512, which is the same setting as in the original paper (Yang et al., 2023). As shown in Table 2, although LieNLSD expands the function library $\Theta$ to include up to second-order terms, while LieGAN's symmetry discovery is limited to a linear search space, LieNLSD still achieves more accurate linear symmetries with fewer parameters.

### 5.4. Application: Guiding Data Augmentation for Neural PDE Solvers

We further apply the symmetries identified by LieNLSD to Lie Point Symmetry Data Augmentation (LPSDA) (Brandstetter et al., 2022) to improve the long rollout accuracy of the Fourier Neural Operator (FNO) (Li et al., 2021), a type of neural PDE solver. Due to the technical limitation of LPSDA to one-dimensional cases, we evaluate it on Burgers' equation, the heat equation (see Appendix I.1), and the KdV equation (see Appendix I.2). The implementation details remain consistent with the original paper (Brandstet-

*Table 4.* Comparison of long rollout test NMSE for FNO without data augmentation, FNO with LPSDA based on Ko et al. (2024), FNO with LPSDA based on LieNLSD, and FNO with LPSDA based on ground truth (GT). The results are presented in the format of mean $\pm$ std over three runs with different random seeds.

| Dataset | FNO + $\emptyset$ | FNO + Ko et al. | FNO + LieNLSD | FNO + GT |
|---|---|---|---|---|
| Burgers' equation | $(2.33 \pm 1.07) \times 10^{-4}$ | $(1.93 \pm 0.75) \times 10^{-4}$ | $(1.80 \pm 0.56) \times 10^{-4}$ | $\mathbf{(1.75 \pm 0.37) \times 10^{-4}}$ |
| Heat equation | $(1.07 \pm 0.10) \times 10^{-1}$ | $(7.33 \pm 1.07) \times 10^{-2}$ | $\mathbf{(5.99 \pm 0.04) \times 10^{-2}}$ | $(6.01 \pm 0.20) \times 10^{-2}$ |
| KdV equation | $(1.74 \pm 0.10) \times 10^{-1}$ | $(1.51 \pm 0.01) \times 10^{-1}$ | $\mathbf{(1.42 \pm 0.16) \times 10^{-1}}$ | $(1.47 \pm 0.02) \times 10^{-1}$ |

ter et al., 2022). As shown in Table 4, compared to FNO without data augmentation, FNO with LPSDA based on LieNLSD improves accuracy by over $20\%$, which is comparable to FNO with LPSDA based on ground truth (GT). This further validates the effectiveness of LieNLSD results.

Additionally, we compare the long rollout accuracy of FNO with LPSDA based on LieNLSD and FNO with LPSDA based on Ko et al. (2024). For Ko et al. (2024), although they cannot obtain explicit expressions for the infinitesimal generators, feeding a given point into their trained MLP still yields the specific values of the infinitesimal generators at that point, thereby guiding data augmentation. The implementation details are consistent with the original paper. As shown in Table 4, Ko et al. (2024) can improve the accuracy of FNO, but not as significantly as our method. In addition to quantitative advantages, Ko et al. (2024) require the explicit expression of the PDE to compute the validity score (see Section 4.2 of their original paper), whereas our LieNLSD does not rely on this prior knowledge.

## 6. Conclusion

To our knowledge, we propose the first pipeline that can determine the number of infinitesimal generators and explicitly provide their expressions containing nonlinear terms. The selection of the function library $\Theta$ is more flexible compared with the Lie algebra representation. This is why the infinitesimal group action solved by LieNLSD can include more nonlinear terms. After training the neural network, we solve for the coefficient matrix $W$ of the infinitesimal group action using mathematical methods. This is why LieNLSD can obtain the entire Lie algebra subspace and determine its dimension. In the future, we expect to use the symmetries identified by LieNLSD to guide the discovery of conserved quantities or differential equations.

## Acknowledgements

Z. Lin was supported by National Key R&D Program of China (2022ZD0160300) and the NSF China (No. 62276004).

## Impact Statement

This paper presents work whose goal is to advance the field of Machine Learning. There are many potential societal consequences of our work, none which we feel must be specifically highlighted here.

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

# A. Preliminary

In this section, we provide an in-depth introduction to some preliminary knowledge of Lie groups and differential equations. For more details, please refer to the textbook (Olver, 1993).

## A.1. Group Actions and Infinitesimal Group Actions

A Lie group $G$ is a mathematical object that is both a smooth manifold and a group. The Lie algebra $\mathfrak{g}$ serves as the tangent space at the identity of the Lie group. A Lie algebra element $\mathbf{v} \in \mathfrak{g}$ can be associated with a Lie group element $g \in G$ through the exponential map $\exp : \mathfrak{g} \to G$, i.e., $g = \exp(\mathbf{v})$. Representing the space of the Lie algebra with a basis $\{\mathbf{v}_1, \mathbf{v}_2, \ldots, \mathbf{v}_r\}$, we have $g = \exp\left(\sum_{i=1}^r \epsilon^i \mathbf{v}_i\right)$, where $\epsilon = (\epsilon^1, \epsilon^2, \ldots, \epsilon^r) \in \mathbb{R}^r$ are the coefficients. The group $G$ acts on a vector space $X = \mathbb{R}^n$ via a group action $\Psi : G \times X \to X$. Correspondingly, we can define the infinitesimal group action of $\mathfrak{g}$ on $X$.

**Definition A.1.** Let $G$ be a Lie group with its corresponding Lie algebra $\mathfrak{g}$. The group action of $G$ on a vector space $X$ is given by $\Psi : G \times X \to X$. Then, **the infinitesimal group action** of $\mathbf{v} \in \mathfrak{g}$ at $x \in X$ is:

$$\psi(\mathbf{v})|_x = \left.\frac{\mathrm{d}}{\mathrm{d}\epsilon}\right|_{\epsilon=0} \Psi(\exp(\epsilon\mathbf{v}), x) \cdot \nabla. \tag{17}$$

We abbreviate $\Psi(g, x)$ and $\psi(\mathbf{v})$ as $g \cdot x$ and $\mathbf{v}$, respectively, when the context is clear. We refer to the infinitesimal group actions of the Lie algebra basis as **infinitesimal generators**.

This is a more general definition that encompasses the nonlinear case. Previous works focusing on linear symmetries (Moskalev et al., 2022; Desai et al., 2022; Yang et al., 2023) commonly use the group representation $\rho_X : G \to \mathrm{GL}(n)$ to characterize group transformations, i.e., $\forall g \in G, x \in X : \Psi(g, x) = \rho_X(g)x$. The corresponding Lie algebra representation $\mathrm{d}\rho_X : \mathfrak{g} \to \mathfrak{gl}(n)$ satisfies $\forall \mathbf{v} \in \mathfrak{g} : \rho_X(\exp(\mathbf{v})) = \exp(\mathrm{d}\rho_X(\mathbf{v}))$. Below, we demonstrate **the relationship between infinitesimal group actions and Lie algebra representations**.

**Proposition A.2.** *Let $G$ be a Lie group with its corresponding Lie algebra $\mathfrak{g}$. Suppose the group action of $G$ on a vector space $X$ is linear:*

$$\forall g \in G, x \in X : \quad \Psi(g, x) = \rho_X(g)x. \tag{18}$$

*Then, we have:*

$$\forall \mathbf{v} \in \mathfrak{g}, x \in X : \quad \psi(\mathbf{v})|_x = \mathrm{d}\rho_X(\mathbf{v})x \cdot \nabla. \tag{19}$$

The proof of Proposition A.2 and a concrete example of group actions and infinitesimal group actions are provided in Appendices B.1 and C.1.

## A.2. Symmetries of Differential Equations

Before defining the symmetries of differential equations, we first explain how groups act on functions. A concrete example is provided in Appendix C.2.

**Definition A.3.** Let $f : X \to U$ be a function, with its graph defined as $\Gamma_f = \{(x, f(x)) : x \in X\} \subset X \times U$. Suppose a Lie group $G$ acts on $X \times U$. Then, the transform of $\Gamma_f$ by $g \in G$ is given by $g \cdot \Gamma_f = \{(\tilde{x}, \tilde{u}) = g \cdot (x, u) : (x, u) \in \Gamma_f\}$. We call a function $\tilde{f} : X \to U$ **the transform of $f$ by $g$**, denoted as $\tilde{f} = g \cdot f$, if its graph satisfies $\Gamma_{\tilde{f}} = g \cdot \Gamma_f$.

Note that $\tilde{f}$ may not always exist, but in this paper, we only consider cases where it does. The solution of a differential equation is expressed in the form of a function. Intuitively, the symmetries of a differential equation describe how one solution can be transformed into another.

**Definition A.4.** Let $\mathscr{S}$ be a system of differential equations, with the independent and dependent variable spaces denoted by $X$ and $U$, respectively. Suppose that a Lie group $G$ acts on $X \times U$. We call $G$ **the symmetry group of** $\mathscr{S}$ if, for any solution $f : X \to U$ of $\mathscr{S}$, and $\forall g \in G$, the transformed function $g \cdot f : X \to U$ is another solution of $\mathscr{S}$.

## A.3. Prolongation

To further study the symmetries of differential equations, we need to "prolong" the group action on the space of independent and dependent variables to the space of derivatives. Given a function $f : X \to U$, where $X = \mathbb{R}^p$ and $U = \mathbb{R}^q$,

it has $q \cdot p_k = q \cdot \binom{p+k-1}{k}$ distinct $k$-th order derivatives $u_J^\alpha = \partial_J f^\alpha(x) = \frac{\partial^k f^\alpha(x)}{\partial x^{j_1} \partial x^{j_2} \dots \partial x^{j_k}}$, where $\alpha \in \{1, \dots, q\}$, $J = (j_1, \dots, j_k)$, and $j_i \in \{1, \dots, p\}$. We denote the space of all $k$-th order derivatives as $U_k = \mathbb{R}^{q \cdot p_k}$ and the space of all derivatives up to order $n$ as $U^{(n)} = U \times U_1 \times \dots U_n = \mathbb{R}^{q \cdot p^{(n)}}$, where $p^{(n)} = \binom{p+n}{n}$. Then, we define the prolongation of $f$.

**Definition A.5.** Let $f : X \to U$ be a smooth function. Then **the $n$-th prolongation of** $f$, $\mathrm{pr}^{(n)} f : X \to U^{(n)}$, is defined as:

$$\mathrm{pr}^{(n)} f(x) = u^{(n)}, \quad u_J^\alpha = \partial_J f^\alpha(x). \tag{20}$$

Based on the above concepts, the $n$-th order system of differential equations $\mathscr{S}$ can be formalized as:

$$F_\nu(x, u^{(n)}) = 0, \quad \nu = 1, \dots, l, \tag{21}$$

where $F : X \times U^{(n)} \to \mathbb{R}^l$. A smooth function $f : X \to U$ is a solution of $\mathscr{S}$ if it satisfies $F(x, \mathrm{pr}^{(n)} f(x)) = 0$.

Below, we explain how to prolong the group action on $X \times U$ to $X \times U^{(n)}$.

**Definition A.6.** Let $G$ be a Lie group acting on the space of independent and dependent variables $X \times U$. Given a point $(x_0, u_0^{(n)}) \in X \times U^{(n)}$, suppose that a smooth function $f : X \to U$ satisfies $u_0^{(n)} = \mathrm{pr}^{(n)} f(x_0)$. Then, **the $n$-th prolongation of** $g \in G$ **at the point** $(x_0, u_0^{(n)})$ is defined as:

$$\mathrm{pr}^{(n)} g \cdot (x_0, u_0^{(n)}) = (\tilde{x}_0, \tilde{u}_0^{(n)}), \tag{22}$$

where

$$(\tilde{x}_0, \tilde{u}_0) = g \cdot (x_0, u_0), \quad \tilde{u}_0^{(n)} = \mathrm{pr}^{(n)}(g \cdot f)(\tilde{x}_0). \tag{23}$$

Note that by the chain rule, the definition of $\mathrm{pr}^{(n)} g$ depends only on $(x_0, u_0^{(n)})$ and is independent of the choice of $f$. Similarly, we can prolong the infinitesimal group action on $X \times U$ to $X \times U^{(n)}$.

**Definition A.7.** Let $G$ be a Lie group acting on $X \times U$, with its corresponding Lie algebra $\mathfrak{g}$. Then, **the $n$-th prolongation of** $\mathbf{v} \in \mathfrak{g}$ **at the point** $(x, u^{(n)}) \in X \times U^{(n)}$ is defined as:

$$\left. \mathrm{pr}^{(n)} \mathbf{v} \right|_{(x, u^{(n)})} = \left. \frac{\mathrm{d}}{\mathrm{d}\epsilon} \right|_{\epsilon=0} \left\{ \mathrm{pr}^{(n)} [\exp(\epsilon \mathbf{v})] \cdot (x, u^{(n)}) \right\} \cdot \nabla. \tag{24}$$

Concrete examples of prolongation are provided in Appendix C.3.

# B. Proof

## B.1. Proof of Proposition A.2

*Proof.* $\forall \mathbf{v} \in \mathfrak{g}, x \in X$, we have:

$$\left. \psi(\mathbf{v}) \right|_x = \left. \frac{\mathrm{d}}{\mathrm{d}\epsilon} \right|_{\epsilon=0} \Psi(\exp(\epsilon \mathbf{v}), x) \cdot \nabla \tag{25}$$

$$= \left. \frac{\mathrm{d}}{\mathrm{d}\epsilon} \right|_{\epsilon=0} \rho_X(\exp(\epsilon \mathbf{v})) x \cdot \nabla \tag{26}$$

$$= \left. \frac{\mathrm{d}}{\mathrm{d}\epsilon} \right|_{\epsilon=0} \exp(\mathrm{d}\rho_X(\epsilon \mathbf{v})) x \cdot \nabla \tag{27}$$

$$= \left. \frac{\mathrm{d}}{\mathrm{d}\epsilon} \right|_{\epsilon=0} \exp(\epsilon \mathrm{d}\rho_X(\mathbf{v})) x \cdot \nabla \tag{28}$$

$$= \mathrm{d}\rho_X(\mathbf{v}) x \cdot \nabla. \tag{29}$$

$\square$

## B.2. Proof of Theorem 4.1

Before proving Theorem 4.1, we first introduce Theorem 2.36 from the textbook (Olver, 1993) as a lemma.

**Lemma B.1.** *Let $G$ be a Lie group acting on $X \times U = \mathbb{R}^p \times \mathbb{R}^q$, with its corresponding Lie algebra $\mathfrak{g}$. Denote the infinitesimal group action of $\mathbf{v} \in \mathfrak{g}$ as:*

$$\mathbf{v} = \sum_{i=1}^{p} \xi^i(x, u) \frac{\partial}{\partial x^i} + \sum_{\alpha=1}^{q} \phi_\alpha(x, u) \frac{\partial}{\partial u^\alpha}, \tag{30}$$

*Then, the $n$-th prolongation of $\mathbf{v}$ is:*

$$\mathrm{pr}^{(n)}\mathbf{v} = \mathbf{v} + \sum_{\alpha=1}^{q} \sum_{J} \phi_\alpha^J(x, u^{(n)}) \frac{\partial}{\partial u_J^\alpha}, \tag{31}$$

*where $J = (j_1, \ldots, j_k)$, with $j_i = 1, \ldots, p$ and $k = 1, \ldots, n$. The coefficients $\phi_\alpha^J$ are given by:*

$$\phi_\alpha^J(x, u^{(n)}) = \mathrm{D}_J \left( \phi_\alpha - \sum_{i=1}^{p} \xi^i u_i^\alpha \right) + \sum_{i=1}^{p} \xi^i u_{J,i}^\alpha, \tag{32}$$

*where $u_i^\alpha = \frac{\partial u^\alpha}{\partial x^i}$, $u_{J,i}^\alpha = \frac{\partial u_J^\alpha}{\partial x^i} = \frac{\partial^{k+1} u^\alpha}{\partial x^i \partial x^{j_1} \ldots \partial x^{j_k}}$, and $\mathrm{D}_J$ represents the total derivative.*

Next, we prove Theorem 4.1 as follows.

*Proof.* By Lemma B.1 and the multivariable derivative formula, we have:

$$\phi_\alpha^J(x, u^{(n)}) = \mathrm{D}_J \left( \Theta^\top W_{p+\alpha} - \sum_{i=1}^{p} \Theta^\top W_i u_i^\alpha \right) + \sum_{i=1}^{p} \Theta^\top W_i u_{J,i}^\alpha \tag{33}$$

$$= \mathrm{D}_J \Theta^\top W_{p+\alpha} - \sum_{i=1}^{p} \sum_{I \subseteq J} u_{I,i}^\alpha \mathrm{D}_{J \setminus I} \Theta^\top W_i + \sum_{i=1}^{p} u_{J,i}^\alpha \Theta^\top W_i \tag{34}$$

$$= -\sum_{i=1}^{p} \sum_{I \subset J} U_{I,i}^\alpha \mathrm{D}_{J \setminus I} \Theta^\top W_i + \mathrm{D}_J \Theta^\top W_{p+\alpha}. \tag{35}$$

□

# C. Example

In Appendices C.1 to C.3, we present several key examples from the textbook (Olver, 1993) to help readers intuitively understand the preliminaries. In Appendix C.4, we provide an example of the construction of $\Theta_n$ in Equation (9).

## C.1. Example of Group Actions and Infinitesimal Group Actions

**Example C.1.** Consider the 2D rotation group $G = \mathrm{SO}(2)$, with its group action given by:

$$\Psi(\epsilon, (x, y)) = (x \cos \epsilon - y \sin \epsilon, x \sin \epsilon + y \cos \epsilon). \tag{36}$$

Then, according to Equation (1) (or Definition A.1), its infinitesimal group action is:

$$\psi(\mathbf{v})|_{(x,y)} = -y \frac{\partial}{\partial x} + x \frac{\partial}{\partial y}. \tag{37}$$

Its Lie algebra representation is:

$$\mathrm{d}\rho_X(\mathbf{v}) = \begin{bmatrix} 0 & -1 \\ 1 & 0 \end{bmatrix}. \tag{38}$$

Thus, Equation (2) (or Proposition A.2) is satisfied.

## C.2. Example of Group Actions on Functions

**Example C.2.** Let $X = \mathbb{R}$ and $U = \mathbb{R}$. The 2D rotation group $G = \mathrm{SO}(2)$ acts on $X \times U$:

$$\epsilon \cdot (x, u) = (x \cos \epsilon - u \sin \epsilon, x \sin \epsilon + u \cos \epsilon). \tag{39}$$

Let $f : X \to U$ be a linear function $u = f(x) = ax + b$. The points on the graph of $f$ transform as follows:

$$(\tilde{x}, \tilde{u}) = (x \cos \epsilon - (ax + b) \sin \epsilon, x \sin \epsilon + (ax + b) \cos \epsilon). \tag{40}$$

To solve $\tilde{u} = \tilde{f}(\tilde{x})$, we first solve for $x$ inversely (assuming $\cos \epsilon - a \sin \epsilon \neq 0$):

$$x = \frac{\tilde{x} + b \sin \epsilon}{\cos \epsilon - a \sin \epsilon}. \tag{41}$$

Then we obtain the transformed function $\tilde{f} = \epsilon \cdot f$:

$$\tilde{u} = \tilde{f}(\tilde{x}) = \frac{\sin \epsilon + a \cos \epsilon}{\cos \epsilon - a \sin \epsilon} \tilde{x} + \frac{b}{\cos \epsilon - a \sin \epsilon}. \tag{42}$$

## C.3. Examples of Prolongation

We first provide an example of the prolongation of a function.

**Example C.3.** Let $X = \mathbb{R}^2$ with coordinates $(x, y)$, and $U = \mathbb{R}$ with coordinate $u$. Then $U_1 = \mathbb{R}^2$ has coordinates $(u_x, u_y)$, $U_2 = \mathbb{R}^3$ has coordinates $(u_{xx}, u_{xy}, u_{yy})$, and $U^{(2)} = U \times U_1 \times U_2 = \mathbb{R}^6$ has coordinates $u^{(2)} = (u; u_x, u_y; u_{xx}, u_{xy}, u_{yy})$.

Consider the function $u = f(x, y)$. Then its second prolongation $u^{(2)} = \mathrm{pr}^{(2)} f(x, y)$ is:

$$(u; u_x, u_y; u_{xx}, u_{xy}, u_{yy}) = \left( f; \frac{\partial f}{\partial x}, \frac{\partial f}{\partial y}; \frac{\partial^2 f}{\partial x^2}, \frac{\partial^2 f}{\partial x \partial y}, \frac{\partial^2 f}{\partial y^2} \right). \tag{43}$$

We next provide an example of the prolongation of a group action.

**Example C.4.** Let $X = \mathbb{R}$ and $U = \mathbb{R}$. The 2D rotation group $G = \mathrm{SO}(2)$ acts on $X \times U$ as shown in Equation (39). Given a point $(x^0, u^0, u_x^0) \in X \times U^{(1)}$, we choose a function $f : X \to U$ as:

$$f(x) = u_x^0 x + (u^0 - u_x^0 x^0), \tag{44}$$

which satisfies:

$$f(x^0) = u^0, \quad f'(x^0) = u_x^0. \tag{45}$$

From Equation (42), the transform of $f$ by $\epsilon$ is (assuming $\cos \epsilon - u_x^0 \sin \epsilon \neq 0$):

$$\tilde{f}(\tilde{x}) = \epsilon \cdot f(\tilde{x}) = \frac{\sin \epsilon + u_x^0 \cos \epsilon}{\cos \epsilon - u_x^0 \sin \epsilon} \tilde{x} + \frac{u^0 - u_x^0 x^0}{\cos \epsilon - u_x^0 \sin \epsilon}. \tag{46}$$

Let $\mathrm{pr}^{(1)} \epsilon \cdot (x^0, u^0, u_x^0) = (\tilde{x}^0, \tilde{u}^0, \tilde{u}_x^0)$. Then, we have:

$$\tilde{u}_x^0 = \tilde{f}'(\tilde{x}^0) = \frac{\sin \epsilon + u_x^0 \cos \epsilon}{\cos \epsilon - u_x^0 \sin \epsilon}. \tag{47}$$

Thus, the first prolongation $\mathrm{pr}^{(1)} \mathrm{SO}(2)$ on $X \times U^{(1)}$ is:

$$\mathrm{pr}^{(1)} \epsilon \cdot (x, u, u_x) = \left( x \cos \epsilon - u \sin \epsilon, x \sin \epsilon + u \cos \epsilon, \frac{\sin \epsilon + u_x \cos \epsilon}{\cos \epsilon - u_x \sin \epsilon} \right). \tag{48}$$

We finally provide an example of the prolongation of an infinitesimal group action.

**Example C.5.** Let $X = \mathbb{R}$ and $U = \mathbb{R}$. The 2D rotation group $G = \mathrm{SO}(2)$ acts on $X \times U$ as shown in Equation (39). Its corresponding infinitesimal group action is:

$$\mathbf{v} = -u\frac{\partial}{\partial x} + x\frac{\partial}{\partial u}. \tag{49}$$

As shown in Equation (48), we have:

$$\mathrm{pr}^{(1)}[\exp(\epsilon\mathbf{v})](x, u, u_x) = \left( x\cos\epsilon - u\sin\epsilon, x\sin\epsilon + u\cos\epsilon, \frac{\sin\epsilon + u_x\cos\epsilon}{\cos\epsilon - u_x\sin\epsilon} \right). \tag{50}$$

Then, according to Equation (3) (or Definition A.7), we obtain the first prolongation of $\mathbf{v}$:

$$\mathrm{pr}^{(1)}\mathbf{v} = -u\frac{\partial}{\partial x} + x\frac{\partial}{\partial u} + (1 + u_x^2)\frac{\partial}{\partial u_x}. \tag{51}$$

### C.4. Example of the Construction of $\Theta_n$

**Example C.6.** Consider the case where $X = \mathbb{R}$ and $U = \mathbb{R}$. In this case, $W = [W_1, W_2]^\top \in \mathbb{R}^{2\times r}$ and $\mathbf{v} = \Theta(x, u)^\top W_1\frac{\partial}{\partial x} + \Theta(x, u)^\top W_2\frac{\partial}{\partial u}$. According to Equations (7) and (8), we have $\mathrm{pr}^{(1)}\mathbf{v} = \mathbf{v} + \phi^x(x, u^{(1)})\frac{\partial}{\partial u_x}$, where $\phi^x(x, u^{(1)}) = -u_x \mathrm{D}_x\Theta^\top W_1 + \mathrm{D}_x\Theta^\top W_2$. This can be rewritten as:

$$\mathrm{pr}^{(1)}\mathbf{v} = \begin{bmatrix} \Theta^\top & 0 \\ 0 & \Theta^\top \\ -u_x\mathrm{D}_x\Theta^\top & \mathrm{D}_x\Theta^\top \end{bmatrix} \begin{bmatrix} W_1 \\ W_2 \end{bmatrix} \cdot \nabla \tag{52}$$

$$= \Theta_1(x, u^{(1)})\mathrm{vec}(W) \cdot \nabla. \tag{53}$$

If we specify the function library $\Theta(x, u) = [1, x, u, x^2, u^2, xu]^\top$, then its total derivative is computed as $\mathrm{D}_x\Theta = [0, 1, u_x, 2x, 2uu_x, u + xu_x]^\top$.

## D. Applications of Symmetry

One application of symmetry is the design of equivariant networks, which embed group symmetries into the network structure such that when the input undergoes a transformation, the output will undergo a corresponding transformation. Cohen & Welling (2016) propose the method of group convolutions, which has been extended to different types of groups (Sosnovik et al., 2021; Worrall & Welling, 2019; Zhu et al., 2022; Naderi et al., 2020; Finzi et al., 2020; MacDonald et al., 2022; Li et al., 2024) and action spaces (Lenssen et al., 2018; Li et al., 2018; Bekkers et al., 2018; Romero et al., 2020; Worrall & Brostow, 2018; Winkels & Cohen, 2019; Esteves et al., 2018; 2019a;b). At the same time, some works construct partial equivariant networks (Wang et al., 2022b) and color equivariant networks (Lengyel et al., 2024) based on the framework of group convolutions. Later, Cohen & Welling (2017) propose the method of steerable convolutions, which is generalized by Weiler & Cesa (2019) to an equivariant convolution framework for the general E(2) group. It has also been applied to different kinds of groups and action spaces (Worrall et al., 2017; Graham et al., 2020; Wang et al., 2022a; Esteves et al., 2020; Li et al., 2025).

Another application of symmetry is the discovery of governing equations (Loiseau & Brunton, 2018; Guan et al., 2021; Yang et al., 2024b), which helps reduce the search space of equations and improve accuracy. In neural PDE solvers, recent works have successfully incorporated symmetries into Physics-Informed Neural Networks (PINNs) (Arora et al., 2024; Lagrave & Tron, 2022; Shumaylov et al., 2024; Zhang et al., 2023; Wang et al., 2025; Akhound-Sadegh et al., 2023) or performed data augmentation based on PDE symmetries (Li et al., 2022; Brandstetter et al., 2022), leading to significant performance improvements. Furthermore, symmetry concepts have enabled notable advances in generative modeling, with a series of works making outstanding contributions to extending diffusion models from Euclidean space to Lie groups (Bertolini et al., 2025; Zhu et al., 2025). However, these works require prior knowledge of symmetries, and incorrect symmetries can have negative effects. Therefore, symmetry discovery from data has become an important topic. We expect to combine our data-driven symmetry discovery approach with these works in the future, eliminating the need for prior knowledge of the symmetry group to guide their processes.

## E. Time and Space Complexity Analysis of LieNLSD

The computational cost of the symmetry discovery procedure is concentrated in the SVD, which has time and space complexities of $\mathcal{O}(mn\min(m,n))$ and $\mathcal{O}(mn)$ for a matrix $A \in \mathbb{R}^{m \times n}$ (Li et al., 2019). When $\frac{M \cdot l}{(p+q) \cdot r} < \epsilon_1$, the time and space complexities of performing SVD on $C \in \mathbb{R}^{(M \cdot l) \times ((p+q) \cdot r)}$ are $\mathcal{O}(M^2 l^2 (p+q)r)$ and $\mathcal{O}(Ml(p+q)r)$. Otherwise, performing SVD on $C^\top C \in \mathbb{R}^{((p+q) \cdot r) \times ((p+q) \cdot r)}$ results in time and space complexities of $\mathcal{O}((p+q)^3 r^3)$ and $\mathcal{O}((p+q)^2 r^2)$. In summary, the time and space complexities of the symmetry discovery procedure are $\mathcal{O}((p+q)r\min(Ml, (p+q)r)^2)$ and $\mathcal{O}((p+q)r\min(Ml, (p+q)r))$, respectively.

## F. Extension of LieNLSD to Linear/Affine Symmetries and Static Data

**Linear/affine symmetry discovery.** LieNLSD can discover a broader range of nonlinear symmetries compared with group representation-based methods (Dehmamy et al., 2021; Moskalev et al., 2022; Yang et al., 2023), thanks to the flexibility in the choice of the function library $\Theta(x, u)$. If we want to restrict LieNLSD to discovering linear symmetries, we can set $\Theta(x, u) = [x, u]^\top \in \mathbb{R}^{(p+q) \times 1}$, where the coefficient matrix $W$ corresponds to the Lie algebra representation according to Equation (2). Furthermore, for affine symmetry discovery, we can include a constant term in the function library: $\Theta(x, u) = [1, x, u]^\top \in \mathbb{R}^{(1+p+q) \times 1}$.

**Symmetry discovery from static data.** Relative to the dynamic data governed by the differential equation $F(x, u^{(n)}) = 0$, we refer to the data governed by the arithmetic equation $F(x) = 0$ as static data. The infinitesimal criterion for the arithmetic equation, similar to Equation (11), is $\mathbf{v}[F(x)] = 0$ whenever $F(x) = 0$. To apply LieNLSD for symmetry discovery from static data, we simply set the prolongation order $n = 0$. In this case, the procedure no longer needs to estimate derivatives but directly uses the original dataset.

## G. Detailed Derivation of Basis Sparsification

We use LADMAP (Lin et al., 2011) to solve the constrained optimization problem presented in Equation (15). The augmented Lagrangian function is constructed as:

$$\mathcal{L}(R, Z, \Lambda) = \|Z\|_{1,1} + \langle \Lambda, QR - Z \rangle + \frac{\beta}{2}\|QR - Z\|_F^2, \tag{54}$$

where $\Lambda \in \mathbb{R}^{((p+q) \cdot r) \times d}$ is the Lagrange multiplier, $\beta > 0$ is the penalty parameter, and the Frobenius norm of a matrix is $\|A\|_F = \sqrt{\sum_{i,j} A_{ij}^2}$. During the iteration process, we alternately update $R$ and $Z$.

We first update $R$ with $Z$ fixed, as shown in Equation (8) of Lin et al. (2011):

$$R_{k+1} = \arg \min_{R^\top R = I} \frac{\beta_k \eta_R}{2} \left\| R - R_k + \frac{Q^\top(\Lambda_k + \beta_k(QR_k - Z_k))}{\beta_k \eta_R} \right\|_F^2. \tag{55}$$

Let $\widetilde{R} = R_k - \frac{Q^\top(\Lambda_k + \beta_k(QR_k - Z_k))}{\beta_k \eta_R}$, then we have:

$$R_{k+1} = \arg \min_{R^\top R = I} \|R - \widetilde{R}\|_F^2 \tag{56}$$

$$= \arg \min_{R^\top R = I} \operatorname{tr}((R - \widetilde{R})^\top (R - \widetilde{R})) \tag{57}$$

$$= \arg \min_{R^\top R = I} \operatorname{tr}(R^\top R) - 2\operatorname{tr}(R^\top \widetilde{R}) + \operatorname{tr}(\widetilde{R}^\top \widetilde{R}) \tag{58}$$

$$= \arg \min_{R^\top R = I} d - 2\operatorname{tr}(R^\top \widetilde{R}) + \operatorname{tr}(\widetilde{R}^\top \widetilde{R}) \tag{59}$$

$$= \arg \max_{R^\top R = I} \operatorname{tr}(R^\top \widetilde{R}). \tag{60}$$

Let $U\Sigma V^\top$ be the SVD of $\widetilde{R} = R_k - \frac{Q^\top(\Lambda_k + \beta_k(QR_k - Z_k))}{\beta_k \eta_R}$, then:

$$R_{k+1} = \arg \max_{R^\top R = I} \operatorname{tr}(R^\top U\Sigma V^\top) \tag{61}$$

$$= \arg \max_{R^\top R = I} \operatorname{tr}(V^\top R^\top U\Sigma). \tag{62}$$

Let $S = V^\top R^\top U$, then $S^\top S = U^\top R V V^\top R^\top U = I$. We have:

$$R_{k+1} = \arg \max_{R^\top R = I} \text{tr}(S\Sigma) \tag{63}$$

$$= \arg \max_{R^\top R = I} \sum_{i=1}^{d} S_{ii}\sigma_i. \tag{64}$$

The orthogonality of $S$ and the non-negativity of the singular values imply that $|S_{ii}| \leq 1$ and $\sigma_i \geq 0$. Therefore, when $S_{ii} = 1$, i.e., when $S = V^\top R^\top U = I$, the expression $\sum_{i=1}^{d} S_{ii}\sigma_i$ reaches its maximum. Due to the orthogonality of $U$ and $V$, we have $R = UV^\top$. Thus, we obtain the update formula for $R$:

$$R_{k+1} = UV^\top. \tag{65}$$

We then update $Z$ with $R$ fixed, as shown in Equation (9) of Lin et al. (2011):

$$Z_{k+1} = \arg \min_Z \|Z\|_{1,1} + \frac{\beta_k \eta_Z}{2} \left\| Z - Z_k - \frac{\Lambda_k + \beta_k(QR_{k+1} - Z_k)}{\beta_k \eta_Z} \right\|_F^2. \tag{66}$$

Let $\widetilde{Z} = Z_k + \frac{\Lambda_k + \beta_k(QR_{k+1} - Z_k)}{\beta_k \eta_Z}$, then:

$$Z_{k+1} = \arg \min_Z \|Z\|_{1,1} + \frac{\beta_k \eta_Z}{2} \|Z - \widetilde{Z}\|_F^2 \tag{67}$$

$$= \arg \min_Z \sum_{i,j} |Z_{ij}| + \frac{\beta_k \eta_Z}{2} \sum_{i,j}(Z_{ij} - \widetilde{Z}_{ij})^2. \tag{68}$$

This problem can be optimized element-wise:

$$(Z_{k+1})_{ij} = \arg \min_z |z| + \frac{\beta_k \eta_Z}{2}(z - \widetilde{Z}_{ij})^2. \tag{69}$$

Let $f(z) = |z| + \frac{\beta_k \eta_Z}{2}(z - \widetilde{Z}_{ij})^2$. When $z > 0$, $f'(z) = 1 + \beta_k \eta_Z(z - \widetilde{Z}_{ij}) = 0$ implies $z = \widetilde{Z}_{ij} - \frac{1}{\beta_k \eta_Z}$, where $\widetilde{Z}_{ij} > \frac{1}{\beta_k \eta_Z}$. When $z < 0$, $f'(z) = -1 + \beta_k \eta_Z(z - \widetilde{Z}_{ij}) = 0$ implies $z = \widetilde{Z}_{ij} + \frac{1}{\beta_k \eta_Z}$, where $\widetilde{Z}_{ij} < -\frac{1}{\beta_k \eta_Z}$. When $z = 0$, if it is the optimal solution, then $0 \in \partial f(0) = [-1 - \beta_k \eta_Z \widetilde{Z}_{ij}, 1 - \beta_k \eta_Z \widetilde{Z}_{ij}]$, and in this case, $-\frac{1}{\beta_k \eta_Z} \leq \widetilde{Z}_{ij} \leq \frac{1}{\beta_k \eta_Z}$. In summary:

$$(Z_{k+1})_{ij} = \begin{cases} \widetilde{Z}_{ij} - \frac{1}{\beta_k \eta_Z}, & \widetilde{Z}_{ij} > \frac{1}{\beta_k \eta_Z}, \\ \widetilde{Z}_{ij} + \frac{1}{\beta_k \eta_Z}, & \widetilde{Z}_{ij} < -\frac{1}{\beta_k \eta_Z}, \\ 0, & -\frac{1}{\beta_k \eta_Z} \leq \widetilde{Z}_{ij} \leq \frac{1}{\beta_k \eta_Z}. \end{cases} \tag{70}$$

Thus, we obtain the update formula for $Z$:

$$Z_{k+1} = \mathcal{S}_{(\beta_k \eta_Z)^{-1}}(\widetilde{Z}) \tag{71}$$

$$= \mathcal{S}_{(\beta_k \eta_Z)^{-1}}\left(Z_k + \frac{\Lambda_k + \beta_k(QR_{k+1} - Z_k)}{\beta_k \eta_Z}\right), \tag{72}$$

where $\mathcal{S}_\epsilon(x) = \text{sgn}(x)\max(|x| - \epsilon, 0)$ is the soft thresholding operator.

## H. Implementation Detail and Visualization Result

### H.1. Linear Symmetry Discovery

**Top quark tagging.** In this task, we observe the four-momenta $\{p_i^\mu\}_{i=1}^{20} \in \mathbb{R}^4$ of the 20 jet constituents with the highest transverse momentum $p_T$, and predict the event label (1 for top quark decay, 0 for other events). We configure an MLP with 3 hidden layers, setting the input dimension to 80, the hidden dimension to 200, and the output dimension to 1. The activation function is ReLU. This is a binary classification task, so we apply the Sigmoid function to the output and use BCE as the loss function. For training, we set the batch size to 256 and use the Adan optimizer (Xie et al., 2024) with a learning

rate of $10^{-3}$. For symmetry discovery, we use Theorem 3 in LieSD (Hu et al., 2025) to ensure that the 20 four-momenta share a group action, which allows us to treat these 20 channels as a single channel in the subsequent analysis. This is a static system, so we set the prolongation order to $n = 0$. With the function library set to $\Theta(p^\mu) = [p^0, p^1, p^2, p^3]^\top \in \mathbb{R}^{4 \times 1}$, Theorem 4.1 gives $\Theta_0$ in Equation (9) as:

$$\Theta_0 = \begin{bmatrix} \Theta^\top & 0 & 0 & 0 \\ 0 & \Theta^\top & 0 & 0 \\ 0 & 0 & \Theta^\top & 0 \\ 0 & 0 & 0 & \Theta^\top \end{bmatrix} \in \mathbb{R}^{4 \times 16}. \tag{73}$$

We sample $M = 100$ points from the training set to construct $C \in \mathbb{R}^{100 \times 16}$ in Equation (13) (in practice, we construct $C^\top C \in \mathbb{R}^{16 \times 16}$). For quantitative analysis, we conduct multiple experiments with different sampled points to report error bars. As shown in Figure 2, we consider the singular values smaller than $\epsilon_2 = 10$ as the effective information of the symmetry group, where the singular values drop sharply to nearly zero. For basis sparsification, we set $\epsilon_1 = 10^{-2}$ and $\epsilon_2 = 10^{-1}$, while keeping the remaining hyperparameters consistent with the original paper (Lin et al., 2011).

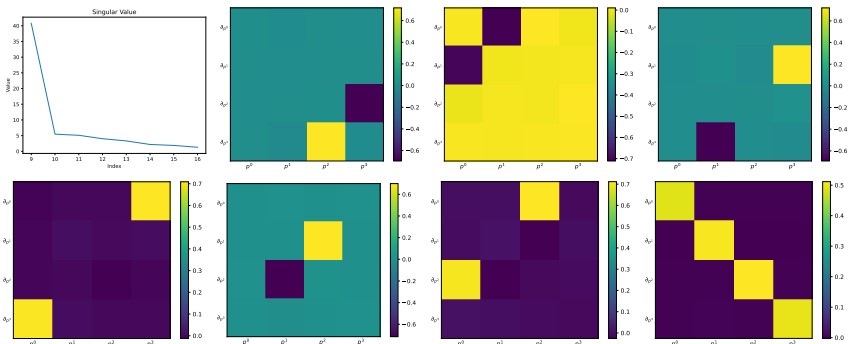

*Figure 2.* Visualization result of symmetry discovery on top quark tagging by LieNLSD. The first subplot shows the last 8 singular values. The other seven subplots display the infinitesimal generators corresponding to the nearly-zero singular values after sparsification.

### H.2. Nonlinear Symmetry Discovery

**Burgers' equation.** For dataset generation, we first uniformly sample $N_x$ spatial points $x \in \mathbb{R}^{N_x}$ on the interval $\left[-\frac{L}{2}, \frac{L}{2}\right]$. By sampling the coefficients of the Fourier series $f(x) = \frac{a_0}{2} + \sum_{n=1}^{N_f} \left(a_n \cos \frac{2n\pi x}{L} + b_n \sin \frac{2n\pi x}{L}\right)$ from a Gaussian distribution, we obtain $N_{ics}$ initial conditions $u_0 \in \mathbb{R}^{N_{ics} \times N_x}$. We then uniformly sample $N_t$ time points $t \in \mathbb{R}^{N_t}$ on the interval $[0, T]$, and numerically integrate the trajectories $u \in \mathbb{R}^{N_{ics} \times N_t \times N_x}$ using the fourth-order Runge-Kutta method (RK4), which, along with $t$ and $x$, forms the discrete dataset. In practice, we set $L = 20$, $N_x = 100$, $N_f = 10$, $N_{ics} = 10$, $T = 2$, and $N_t = 1000$.

For LieNLSD, we set the prolongation order to $n = 2$, and use the central difference method to estimate the derivatives $u_t, u_x, u_{xx}, u_{tx}$. We configure an MLP with 3 hidden layers to fit the mapping $u_t = f(u, u_x, u_{xx})$, setting the input dimension to 3, the hidden dimension to 200, and the output dimension to 1. The activation function is Sigmoid. For training, we set the batch size to 256 and use the Adan optimizer (Xie et al., 2024) with a learning rate of $10^{-3}$. For symmetry discovery, we specify the function library up to second-order terms as $\Theta(t, x, u) = [1, t, x, u, t^2, x^2, u^2, tx, tu, xu]^\top \in \mathbb{R}^{10 \times 1}$. The infinitesimal group action $\mathbf{v}$ has the form:

$$\mathbf{v} = \Theta(t, x, u)^\top W_1 \frac{\partial}{\partial t} + \Theta(t, x, u)^\top W_2 \frac{\partial}{\partial x} + \Theta(t, x, u)^\top W_3 \frac{\partial}{\partial u}, \tag{74}$$

where $W = [W_1, W_2, W_3]^\top \in \mathbb{R}^{3 \times 10}$. According to Theorem 4.1, the second prolongation of $\mathbf{v}$ is:

$$
\begin{aligned}
\mathrm{pr}^{(2)}\mathbf{v} =& \Theta^\top W_3 \frac{\partial}{\partial u} \\
&+ (-u_t \mathrm{D}_x \Theta^\top W_1 - u_x \mathrm{D}_x \Theta^\top W_2 + \mathrm{D}_x \Theta^\top W_3) \frac{\partial}{\partial u_x} \\
&+ [-(u_t \mathrm{D}_{xx} \Theta^\top + 2u_{tx} \mathrm{D}_x \Theta^\top) W_1 - (u_x \mathrm{D}_{xx} \Theta^\top + 2u_{xx} \mathrm{D}_x \Theta^\top) W_2 + \mathrm{D}_{xx} \Theta^\top W_3] \frac{\partial}{\partial u_{xx}} \\
&+ (-u_t \mathrm{D}_t \Theta^\top W_1 - u_x \mathrm{D}_t \Theta^\top W_2 + \mathrm{D}_t \Theta^\top W_3) \frac{\partial}{\partial u_t} \\
&+ \cdots,
\end{aligned}
\tag{75}
$$

where $\mathrm{D}_t\Theta = [0, 1, 0, u_t, 2t, 0, 2uu_t, x, u + tu_t, xu_t]^\top$, $\mathrm{D}_x\Theta = [0, 0, 1, u_x, 0, 2x, 2uu_x, t, tu_x, u + xu_x]^\top$, and $\mathrm{D}_{xx}\Theta = [0, 0, 0, u_{xx}, 0, 2, 2(u_x^2 + uu_{xx}), 0, tu_{xx}, 2u_x + xu_{xx}]^\top$. We omit the irrelevant terms here because we assume the form of the differential equation is $F(u, u_x, u_{xx}, u_t) = f(u, u_x, u_{xx}) - u_t = 0$, which only depends on $u, u_x, u_{xx}, u_t$. More specifically, for example, since we have $\frac{\partial F}{\partial u_{tx}} = 0$, the row corresponding to $\frac{\partial}{\partial u_{tx}}$ in $\Theta_2$ of Equation (12) will be ignored. Equation (75) gives $\Theta_2$ in Equation (9) as:

$$
\Theta_2 = \begin{bmatrix}
0 & 0 & \Theta^\top \\
-u_t \mathrm{D}_x \Theta^\top & -u_x \mathrm{D}_x \Theta^\top & \mathrm{D}_x \Theta^\top \\
-(u_t \mathrm{D}_{xx} \Theta^\top + 2u_{tx} \mathrm{D}_x \Theta^\top) & -(u_x \mathrm{D}_{xx} \Theta^\top + 2u_{xx} \mathrm{D}_x \Theta^\top) & \mathrm{D}_{xx} \Theta^\top \\
-u_t \mathrm{D}_t \Theta^\top & -u_x \mathrm{D}_t \Theta^\top & \mathrm{D}_t \Theta^\top
\end{bmatrix} \in \mathbb{R}^{4 \times 30}.
\tag{76}
$$

We sample $M = 100$ points from the training set to construct $C \in \mathbb{R}^{100 \times 30}$ in Equation (13) (in practice, we construct $C^\top C \in \mathbb{R}^{30 \times 30}$). For quantitative analysis, we conduct multiple experiments with different sampled points to report error bars. As shown in Figure 3, we consider the singular values smaller than $\epsilon_2 = 0.5$ as the effective information of the symmetry group, where the singular values drop sharply to nearly zero. For basis sparsification, we set $\epsilon_1 = 10^{-4}$ and $\epsilon_2 = 10^{-4}$, while keeping the remaining hyperparameters consistent with the original paper (Lin et al., 2011).

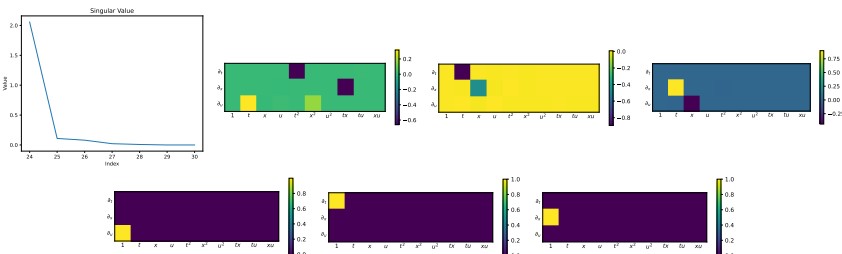

*Figure 3.* Visualization result of symmetry discovery on Burgers' equation by LieNLSD. The first subplot shows the last 7 singular values. The other six subplots display the infinitesimal generators corresponding to the nearly-zero singular values after sparsification.

**Wave equation.** For dataset generation, we first rewrite the equation in its first-order form:

$$
\begin{cases}
u_t = v, \\
v_t = u_{xx} + u_{yy}.
\end{cases}
\tag{77}
$$

We uniformly sample $N_x^2$ spatial points on the interval $\left[-\frac{L}{2}, \frac{L}{2}\right]^2$, with coordinates $x \in \mathbb{R}^{N_x^2}$ and $y \in \mathbb{R}^{N_x^2}$. By sampling the coefficients of the Fourier series $f(x, y) = \frac{a_0}{4} + \sum_{m=1}^{N_f} \sum_{n=1}^{N_f} \left(a_{mn} \cos \frac{2m\pi x}{L} \cos \frac{2n\pi y}{L} + b_{mn} \cos \frac{2m\pi x}{L} \sin \frac{2n\pi y}{L} + c_{mn} \sin \frac{2m\pi x}{L} \cos \frac{2n\pi y}{L} + d_{mn} \sin \frac{2m\pi x}{L} \sin \frac{2n\pi y}{L}\right)$ from a Gaussian distribution, we obtain $N_{ics}$ initial conditions $u_0 \in \mathbb{R}^{N_{ics} \times N_x^2}$ and $v_0 \in \mathbb{R}^{N_{ics} \times N_x^2}$. We then uniformly sample $N_t$ time points $t \in \mathbb{R}^{N_t}$ on the interval $[0, T]$, and numerically integrate the trajectories $u \in \mathbb{R}^{N_{ics} \times N_t \times N_x^2}$ and $v \in \mathbb{R}^{N_{ics} \times N_t \times N_x^2}$ using the fourth-order Runge-Kutta method (RK4). We sample every 10 points in space to obtain the discrete dataset $t \in \mathbb{R}^{N_t}, \widetilde{x} \in \mathbb{R}^{(N_x/10)^2}, \widetilde{y} \in \mathbb{R}^{(N_x/10)^2}, \widetilde{u} \in \mathbb{R}^{N_{ics} \times N_t \times (N_x/10)^2}$. In practice, we set $L = 20$, $N_x = 100$, $N_f = 3$, $N_{ics} = 10$, $T = 2$, and $N_t = 1000$.

For LieNLSD, we set the prolongation order to $n = 2$, and use the central difference method to estimate the derivatives $u_t, u_x, u_y, u_{tt}, u_{xx}, u_{yy}, u_{tx}, u_{ty}, u_{xy}$. We configure an MLP with 3 hidden layers to fit the mapping $u_{tt} = f(u, u_x, u_y, u_{xx}, u_{yy}, u_{xy})$, setting the input dimension to 6, the hidden dimension to 200, and the output dimension to 1. The activation function is Sigmoid. For training, we set the batch size to 256 and use the Adan optimizer (Xie et al., 2024) with a learning rate of $10^{-3}$. For symmetry discovery, we specify the function library up to second-order terms as $\Theta(t, x, y, u) = [1, t, x, y, u, t^2, x^2, y^2, u^2, tx, ty, tu, xy, xu, yu]^\top \in \mathbb{R}^{15 \times 1}$. The infinitesimal group action $\mathbf{v}$ has the form:

$$\mathbf{v} = \Theta(t, x, y, u)^\top W_1 \frac{\partial}{\partial t} + \Theta(t, x, y, u)^\top W_2 \frac{\partial}{\partial x} + \Theta(t, x, y, u)^\top W_3 \frac{\partial}{\partial y} + \Theta(t, x, y, u)^\top W_4 \frac{\partial}{\partial u}, \tag{78}$$

where $W = [W_1, W_2, W_3, W_4]^\top \in \mathbb{R}^{4 \times 15}$. According to Theorem 4.1, the second prolongation of $\mathbf{v}$ is:

$$
\begin{aligned}
\mathrm{pr}^{(2)} \mathbf{v} =& \Theta^\top W_4 \frac{\partial}{\partial u} \\
&+ (-u_t \mathrm{D}_x \Theta^\top W_1 - u_x \mathrm{D}_x \Theta^\top W_2 - u_y \mathrm{D}_x \Theta^\top W_3 + \mathrm{D}_x \Theta^\top W_4) \frac{\partial}{\partial u_x} \\
&+ (-u_t \mathrm{D}_y \Theta^\top W_1 - u_x \mathrm{D}_y \Theta^\top W_2 - u_y \mathrm{D}_y \Theta^\top W_3 + \mathrm{D}_y \Theta^\top W_4) \frac{\partial}{\partial u_y} \\
&+ [-(u_t \mathrm{D}_{xx} \Theta^\top + 2 u_{tx} \mathrm{D}_x \Theta^\top) W_1 - (u_x \mathrm{D}_{xx} \Theta^\top + 2 u_{xx} \mathrm{D}_x \Theta^\top) W_2 \\
&\quad - (u_y \mathrm{D}_{xx} \Theta^\top + 2 u_{xy} \mathrm{D}_x \Theta^\top) W_3 + \mathrm{D}_{xx} \Theta^\top W_4] \frac{\partial}{\partial u_{xx}} \\
&+ [-(u_t \mathrm{D}_{yy} \Theta^\top + 2 u_{ty} \mathrm{D}_y \Theta^\top) W_1 - (u_x \mathrm{D}_{yy} \Theta^\top + 2 u_{xy} \mathrm{D}_y \Theta^\top) W_2 \\
&\quad - (u_y \mathrm{D}_{yy} \Theta^\top + 2 u_{yy} \mathrm{D}_y \Theta^\top) W_3 + \mathrm{D}_{yy} \Theta^\top W_4] \frac{\partial}{\partial u_{yy}} \\
&+ [-(u_t \mathrm{D}_{xy} \Theta^\top + u_{tx} \mathrm{D}_y \Theta^\top + u_{ty} \mathrm{D}_x \Theta^\top) W_1 - (u_x \mathrm{D}_{xy} \Theta^\top + u_{xx} \mathrm{D}_y \Theta^\top + u_{xy} \mathrm{D}_x \Theta^\top) W_2 \\
&\quad - (u_y \mathrm{D}_{xy} \Theta^\top + u_{xy} \mathrm{D}_y \Theta^\top + u_{yy} \mathrm{D}_x \Theta^\top) W_3 + \mathrm{D}_{xy} \Theta^\top W_4] \frac{\partial}{\partial u_{xy}} \\
&+ [-(u_t \mathrm{D}_{tt} \Theta^\top + 2 u_{tt} \mathrm{D}_t \Theta^\top) W_1 - (u_x \mathrm{D}_{tt} \Theta^\top + 2 u_{tx} \mathrm{D}_t \Theta^\top) W_2 \\
&\quad - (u_y \mathrm{D}_{tt} \Theta^\top + 2 u_{ty} \mathrm{D}_t \Theta^\top) W_3 + \mathrm{D}_{tt} \Theta^\top W_4] \frac{\partial}{\partial u_{tt}} \\
&+ \ldots,
\end{aligned}
\tag{79}
$$

where

$$
\begin{cases}
\mathrm{D}_t \Theta = [0, 1, 0, 0, u_t, 2t, 0, 0, 2u u_t, x, y, u + t u_t, 0, x u_t, y u_t]^\top, \\
\mathrm{D}_x \Theta = [0, 0, 1, 0, u_x, 0, 2x, 0, 2u u_x, t, 0, t u_x, y, u + x u_x, y u_x]^\top, \\
\mathrm{D}_y \Theta = [0, 0, 0, 1, u_y, 0, 0, 2y, 2u u_y, 0, t, t u_y, x, x u_y, u + y u_y]^\top, \\
\mathrm{D}_{tt} \Theta = [0, 0, 0, 0, u_{tt}, 2, 0, 0, 2(u_t^2 + u u_{tt}), 0, 0, 2u_t + t u_{tt}, 0, x u_{tt}, y u_{tt}]^\top, \\
\mathrm{D}_{xx} \Theta = [0, 0, 0, 0, u_{xx}, 0, 2, 0, 2(u_x^2 + u u_{xx}), 0, 0, t u_{xx}, 0, 2u_x + x u_{xx}, y u_{xx}]^\top, \\
\mathrm{D}_{yy} \Theta = [0, 0, 0, 0, u_{yy}, 0, 0, 2, 2(u_y^2 + u u_{yy}), 0, 0, t u_{yy}, 0, x u_{yy}, 2u_y + y u_{yy}]^\top, \\
\mathrm{D}_{xy} \Theta = [0, 0, 0, 0, u_{xy}, 0, 0, 0, 2(u_x u_y + u u_{xy}), 0, 0, t u_{xy}, 1, u_y + x u_{xy}, u_x + y u_{xy}]^\top.
\end{cases}
\tag{80}
$$

It gives $\Theta_2$ in Equation (9) as:

$$
\Theta_2 = \begin{bmatrix}
0 & 0 & 0 & \Theta^\top \\
-u_t \mathrm{D}_x \Theta^\top & -u_x \mathrm{D}_x \Theta^\top & -u_y \mathrm{D}_x \Theta^\top & \mathrm{D}_x \Theta^\top \\
-u_t \mathrm{D}_y \Theta^\top & -u_x \mathrm{D}_y \Theta^\top & -u_y \mathrm{D}_y \Theta^\top & \mathrm{D}_y \Theta^\top \\
-(u_t \mathrm{D}_{xx} \Theta^\top + 2 u_{tx} \mathrm{D}_x \Theta^\top) & -(u_x \mathrm{D}_{xx} \Theta^\top + 2 u_{xx} \mathrm{D}_x \Theta^\top) & -(u_y \mathrm{D}_{xx} \Theta^\top + 2 u_{xy} \mathrm{D}_x \Theta^\top) & \mathrm{D}_{xx} \Theta^\top \\
-(u_t \mathrm{D}_{yy} \Theta^\top + 2 u_{ty} \mathrm{D}_y \Theta^\top) & -(u_x \mathrm{D}_{yy} \Theta^\top + 2 u_{xy} \mathrm{D}_y \Theta^\top) & -(u_y \mathrm{D}_{yy} \Theta^\top + 2 u_{yy} \mathrm{D}_y \Theta^\top) & \mathrm{D}_{yy} \Theta^\top \\
-(u_t \mathrm{D}_{xy} \Theta^\top + u_{tx} \mathrm{D}_y \Theta^\top + u_{ty} \mathrm{D}_x \Theta^\top) & -(u_x \mathrm{D}_{xy} \Theta^\top + u_{xx} \mathrm{D}_y \Theta^\top + u_{xy} \mathrm{D}_x \Theta^\top) & -(u_y \mathrm{D}_{xy} \Theta^\top + u_{xy} \mathrm{D}_y \Theta^\top + u_{yy} \mathrm{D}_x \Theta^\top) & \mathrm{D}_{xy} \Theta^\top \\
-(u_t \mathrm{D}_{tt} \Theta^\top + 2 u_{tt} \mathrm{D}_t \Theta^\top) & -(u_x \mathrm{D}_{tt} \Theta^\top + 2 u_{tx} \mathrm{D}_t \Theta^\top) & -(u_y \mathrm{D}_{tt} \Theta^\top + 2 u_{ty} \mathrm{D}_t \Theta^\top) & \mathrm{D}_{tt} \Theta^\top
\end{bmatrix} \in \mathbb{R}^{7 \times 60}.
\tag{81}
$$

We sample $M = 100$ points from the training set to construct $C \in \mathbb{R}^{100 \times 60}$ in Equation (13) (in practice, we construct $C^\top C \in \mathbb{R}^{60 \times 60}$). For quantitative analysis, we conduct multiple experiments with different sampled points to report error bars. As shown in Figure 4, we consider the singular values smaller than $\epsilon_2 = 1$ as the effective information of the symmetry

group, where the singular values drop sharply to nearly zero. For basis sparsification, we set $\epsilon_1 = 10^{-4}$ and $\epsilon_2 = 10^{-3}$, while keeping the remaining hyperparameters consistent with the original paper (Lin et al., 2011).

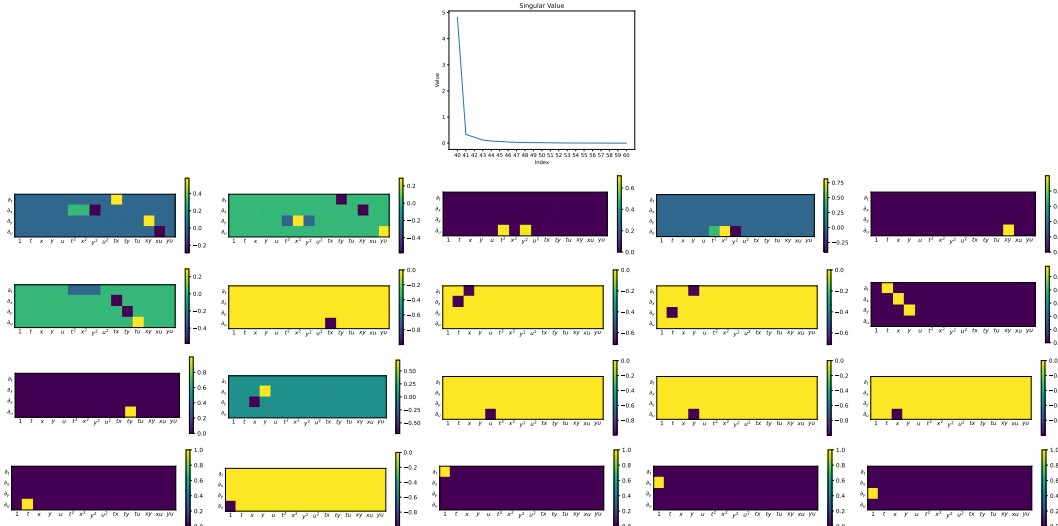

*Figure 4.* Visualization result of symmetry discovery on the wave equation by LieNLSD. The first subplot shows the last 21 singular values. The other twenty subplots display the infinitesimal generators corresponding to the nearly-zero singular values after sparsification.

**Schrödinger equation.** For dataset generation, we follow the same approach as the wave equation to obtain the time series $t \in \mathbb{R}^{N_t}$, grid coordinates $x, y \in \mathbb{R}^{N_x^2}$, and trajectories $u, v \in \mathbb{R}^{N_{ics} \times N_t \times N_x^2}$. We sample every 10 points in space to get the discrete dataset $t \in \mathbb{R}^{N_t}, \widetilde{x} \in \mathbb{R}^{(N_x/10)^2}, \widetilde{y} \in \mathbb{R}^{(N_x/10)^2}, \widetilde{u} \in \mathbb{R}^{N_{ics} \times N_t \times (N_x/10)^2}, \widetilde{v} \in \mathbb{R}^{N_{ics} \times N_t \times (N_x/10)^2}$. For the parameters, we set $L = 20$, $N_x = 100$, $N_f = 2$, $N_{ics} = 10$, $T = 2$, and $N_t = 1000$.

For LieNLSD, we set the prolongation order to $n = 2$, and use the central difference method to estimate the derivatives $u_t, u_x, u_y, u_{xx}, u_{yy}, u_{tx}, u_{ty}, u_{xy}, v_t, v_x, v_y, v_{xx}, v_{yy}, v_{tx}, v_{ty}, v_{xy}$. We configure an MLP with 3 hidden layers to fit the mapping $(u_t, v_t) = \mathbf{f}(u, u_x, u_y, u_{xx}, u_{yy}, u_{xy}, v, v_x, v_y, v_{xx}, v_{yy}, v_{xy})$, setting the input dimension to 12, the hidden dimension to 200, and the output dimension to 2. The activation function is Sigmoid. For training, we set the batch size to 256 and use the Adan optimizer (Xie et al., 2024) with a learning rate of $10^{-3}$. For symmetry discovery, we specify the function library up to second-order terms as $\Theta(t, x, y, u, v) = [1, t, x, y, u, v, t^2, x^2, y^2, u^2, v^2, tx, ty, tu, tv, xy, xu, xv, yu, yv, uv]^\top \in \mathbb{R}^{21 \times 1}$. The infinitesimal group action $\mathbf{v}$ has the form:

$$\mathbf{v} = \Theta(t,x,y,u,v)^\top W_1 \frac{\partial}{\partial t} + \Theta(t,x,y,u,v)^\top W_2 \frac{\partial}{\partial x} + \Theta(t,x,y,u,v)^\top W_3 \frac{\partial}{\partial y} + \Theta(t,x,y,u,v)^\top W_4 \frac{\partial}{\partial u} + \Theta(t,x,y,u,v)^\top W_5 \frac{\partial}{\partial v},$$
(82)

where $W = [W_1, W_2, W_3, W_4, W_5]^\top \in \mathbb{R}^{5 \times 21}$. Similarly to the wave equation, Theorem 4.1 gives $\Theta_2$ in Equation (9) as:

$$\Theta_2 = \begin{bmatrix} 0 & 0 & 0 & \Theta^\top & 0 \\ -u_t D_x \Theta^\top & -u_x D_x \Theta^\top & -u_y D_x \Theta^\top & D_x \Theta^\top & 0 \\ -u_t D_y \Theta^\top & -u_x D_y \Theta^\top & -u_y D_y \Theta^\top & D_y \Theta^\top & 0 \\ -(u_t D_{xx} \Theta^\top + 2u_{tx} D_x \Theta^\top) & -(u_x D_{xx} \Theta^\top + 2u_{xx} D_x \Theta^\top) & -(u_y D_{xx} \Theta^\top + 2u_{xy} D_x \Theta^\top) & D_{xx} \Theta^\top & 0 \\ -(u_t D_{yy} \Theta^\top + 2u_{ty} D_y \Theta^\top) & -(u_x D_{yy} \Theta^\top + 2u_{xy} D_y \Theta^\top) & -(u_y D_{yy} \Theta^\top + 2u_{yy} D_y \Theta^\top) & D_{yy} \Theta^\top & 0 \\ -(u_t D_{xy} \Theta^\top + u_{tx} D_y \Theta^\top + u_{ty} D_x \Theta^\top) & -(u_x D_{xy} \Theta^\top + u_{xx} D_y \Theta^\top + u_{xy} D_x \Theta^\top) & -(u_y D_{xy} \Theta^\top + u_{xy} D_y \Theta^\top + u_{yy} D_x \Theta^\top) & D_{xy} \Theta^\top & 0 \\ 0 & 0 & 0 & 0 & \Theta^\top \\ -v_t D_x \Theta^\top & -v_x D_x \Theta^\top & -v_y D_x \Theta^\top & 0 & D_x \Theta^\top \\ -v_t D_y \Theta^\top & -v_x D_y \Theta^\top & -v_y D_y \Theta^\top & 0 & D_y \Theta^\top \\ -(v_t D_{xx} \Theta^\top + 2v_{tx} D_x \Theta^\top) & -(v_x D_{xx} \Theta^\top + 2v_{xx} D_x \Theta^\top) & -(v_y D_{xx} \Theta^\top + 2v_{xy} D_x \Theta^\top) & 0 & D_{xx} \Theta^\top \\ -(v_t D_{yy} \Theta^\top + 2v_{ty} D_y \Theta^\top) & -(v_x D_{yy} \Theta^\top + 2v_{xy} D_y \Theta^\top) & -(v_y D_{yy} \Theta^\top + 2v_{yy} D_y \Theta^\top) & 0 & D_{yy} \Theta^\top \\ -(v_t D_{xy} \Theta^\top + v_{tx} D_y \Theta^\top + v_{ty} D_x \Theta^\top) & -(v_x D_{xy} \Theta^\top + v_{xx} D_y \Theta^\top + v_{xy} D_x \Theta^\top) & -(v_y D_{xy} \Theta^\top + v_{xy} D_y \Theta^\top + v_{yy} D_x \Theta^\top) & 0 & D_{xy} \Theta^\top \\ -u_t D_t \Theta^\top & -u_x D_t \Theta^\top & -u_y D_t \Theta^\top & D_t \Theta^\top & 0 \\ -v_t D_t \Theta^\top & -v_x D_t \Theta^\top & -v_y D_t \Theta^\top & 0 & D_t \Theta^\top \end{bmatrix} \in \mathbb{R}^{14 \times 105}.$$
(83)

We sample $M = 100$ points from the training set to construct $C \in \mathbb{R}^{100 \times 105}$ in Equation (13). For quantitative analysis, we conduct multiple experiments with different sampled points to report error bars. As shown in Figure 5, we consider the singular values smaller than $\epsilon_2 = 2$ as the effective information of the symmetry group, where the singular values

drop sharply to nearly zero. For basis sparsification, we set $\epsilon_1 = 10^{-4}$ and $\epsilon_2 = 10^{-3}$, while keeping the remaining hyperparameters consistent with the original paper (Lin et al., 2011).

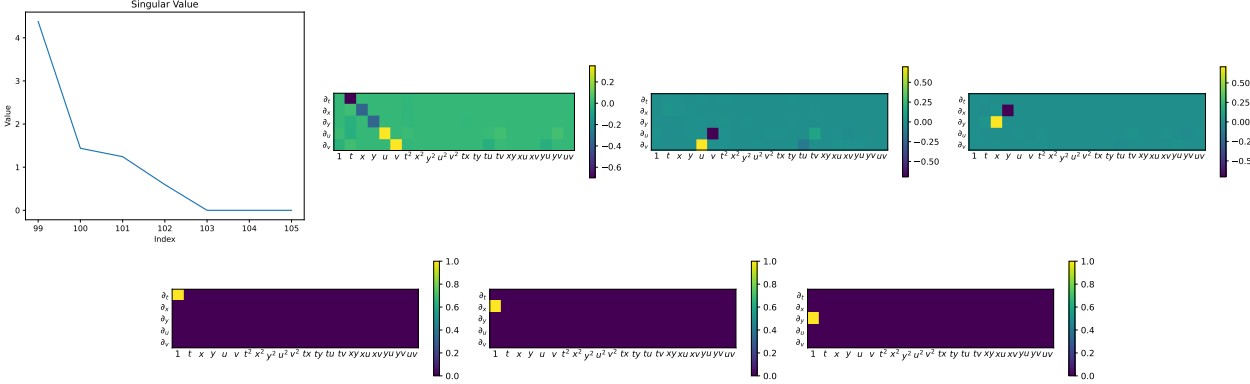

*Figure 5.* Visualization result of symmetry discovery on the Schrödinger equation by LieNLSD. The first subplot shows the last 7 singular values. The other six subplots display the infinitesimal generators corresponding to the nearly-zero singular values after sparsification.

# I. Additional Experiment

We supplement experiments on dynamic data governed by the heat equation, the Korteweg-De Vries (KdV) equation, and reaction-diffusion system. The quantitative comparison between LieNLSD and LieGAN on additional experiments is provided in Table 5.

*Table 5.* Quantitative comparison of LieNLSD and LieGAN on additional experiments. The Grassmann distance is presented in the format of mean $\pm$ std over three runs.

| Dataset | Model | Grassmann distance ($\downarrow$) | Parameters |
|---|---|---|---|
| Heat equation | LieNLSD | $(\mathbf{7.39 \pm 1.04}) \times \mathbf{10^{-4}}$ | 81K |
| | LieGAN | $2.59 \pm 0.04$ | 265K |
| KdV equation | LieNLSD | $(\mathbf{5.24 \pm 1.55}) \times \mathbf{10^{-3}}$ | 82K |
| | LieGAN | $1.56 \pm 0.00$ | 265K |
| Reaction-diffusion system | LieNLSD | $(\mathbf{1.24 \pm 0.18}) \times \mathbf{10^{-1}}$ | 83K |
| | LieGAN | $1.11 \pm 0.11$ | 266K |

*Table 6.* Infinitesimal generators found on additional experiments by LieNLSD.

| Dataset | Heat equation | KdV equation | Reaction-diffusion system |
|---|---|---|---|
| Generators | $\mathbf{v}_1 = (2t + x^2)\dfrac{\partial}{\partial u}, \quad \mathbf{v}_2 = 2t\dfrac{\partial}{\partial t} + x\dfrac{\partial}{\partial x},$ $\mathbf{v}_3 = 2t\dfrac{\partial}{\partial x} - xu\dfrac{\partial}{\partial u}, \quad \mathbf{v}_4 = x\dfrac{\partial}{\partial u}, \quad \mathbf{v}_5 = u\dfrac{\partial}{\partial u},$ $\mathbf{v}_6 = \dfrac{\partial}{\partial u}, \quad \mathbf{v}_7 = \dfrac{\partial}{\partial x}, \quad \mathbf{v}_8 = \dfrac{\partial}{\partial t}$ | $\mathbf{v}_1 = -3t\dfrac{\partial}{\partial t} - x\dfrac{\partial}{\partial x} + 2u\dfrac{\partial}{\partial u},$ $\mathbf{v}_2 = t\dfrac{\partial}{\partial x} + \dfrac{\partial}{\partial u},$ $\mathbf{v}_3 = \dfrac{\partial}{\partial t}, \quad \mathbf{v}_4 = \dfrac{\partial}{\partial x}$ | $\mathbf{v}_1 = -v\dfrac{\partial}{\partial u} + u\dfrac{\partial}{\partial v},$ $\mathbf{v}_2 = -y\dfrac{\partial}{\partial x} + x\dfrac{\partial}{\partial y},$ $\mathbf{v}_3 = \dfrac{\partial}{\partial t}, \quad \mathbf{v}_4 = \dfrac{\partial}{\partial x}, \quad \mathbf{v}_5 = \dfrac{\partial}{\partial y}$ |

## I.1. Heat Equation

The heat equation describes the evolution of the temperature distribution over time in a given medium. Its one-dimensional form is:

$$u_t = u_{xx}, \tag{84}$$

where $u(t, x)$ is the temperature at time $t$ and position $x$. The experimental setup is consistent with Burgers' equation.

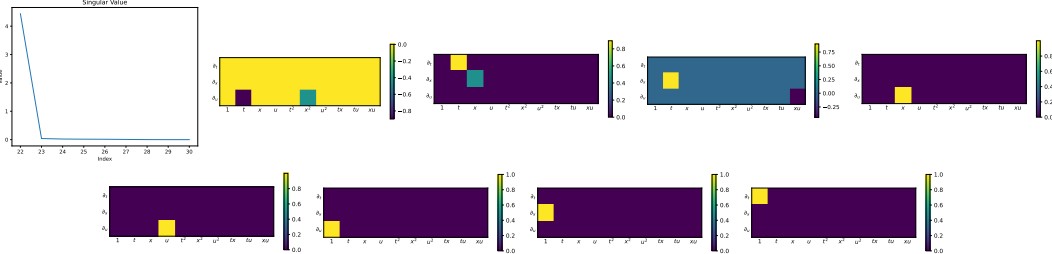

*Figure 6.* Visualization result of symmetry discovery on the heat equation by LieNLSD. The first subplot shows the last 9 singular values. The other eight subplots display the infinitesimal generators corresponding to the nearly-zero singular values after sparsification.

We present the visualization result of LieNLSD on the heat equation in Figure 6. LieNLSD obtains 8 nearly zero singular values, which indicates that the number of infinitesimal generators is 8. The corresponding explicit expressions are shown in Table 6. The group actions they generate for the symmetry group are $\exp(\epsilon \mathbf{v}_1)(t, x, u) = (t, x, u + \epsilon(2t + x^2))$, $\exp(\epsilon \mathbf{v}_2)(t, x, u) = (e^{2\epsilon}t, e^{\epsilon}x, u)$, $\exp(\epsilon \mathbf{v}_3)(t, x, u) = (t, x + 2\epsilon t, ue^{-\epsilon x - \epsilon^2 t})$, $\exp(\epsilon \mathbf{v}_4)(t, x, u) = (t, x, u + \epsilon x)$, $\exp(\epsilon \mathbf{v}_5)(t, x, u) = (t, x, e^{\epsilon}u)$, $\exp(\epsilon \mathbf{v}_6)(t, x, u) = (t, x, u + \epsilon)$, $\exp(\epsilon \mathbf{v}_7)(t, x, u) = (t, x + \epsilon, u)$, and $\exp(\epsilon \mathbf{v}_8)(t, x, u) = (t + \epsilon, x, u)$. The practical meaning is that, if $u = f(t, x)$ is a solution to the heat equation, then $u_1 = f(t, x) + \epsilon(2t + x^2)$, $u_2 = f(e^{-2\epsilon}t, e^{-\epsilon}x)$, $u_3 = e^{-\epsilon x + \epsilon^2 t}f(t, x - 2\epsilon t)$, $u_4 = f(t, x) + \epsilon x$, $u_5 = e^{\epsilon}f(t, x)$, $u_6 = f(t, x) + \epsilon$, $u_7 = f(t, x - \epsilon)$, and $u_8 = f(t - \epsilon, x)$ are also solutions.

## I.2. Korteweg-De Vries (KdV) Equation

The KdV equation describes the propagation of solitary waves on shallow water surfaces. Its standard form is:

$$u_t + u_{xxx} + uu_x = 0, \tag{85}$$

where $u(t, x)$ is the wave profile at time $t$ and position $x$. The experimental setup is consistent with Burgers' equation.

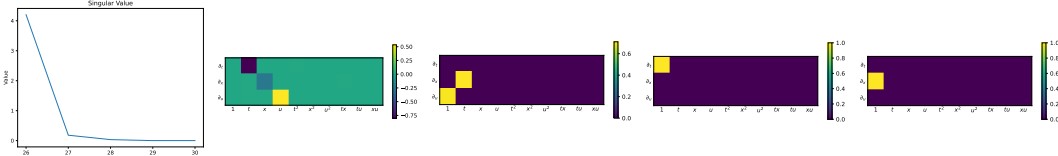

*Figure 7.* Visualization result of symmetry discovery on the KdV equation by LieNLSD. The first subplot shows the last 5 singular values. The other four subplots display the infinitesimal generators corresponding to the nearly-zero singular values after sparsification.

We present the visualization result of LieNLSD on the KdV equation in Figure 7. LieNLSD obtains 4 nearly zero singular values, which indicates that the number of infinitesimal generators is 4. The corresponding explicit expressions are shown in Table 6. Then, if $u = f(t, x)$ is a solution to the KdV equation, we can obtain several derived solutions through these infinitesimal generators. Specifically, $\mathbf{v}_1$ represents scaling $u_1 = e^{2\epsilon}f(e^{3\epsilon}t, e^{\epsilon}x)$, $\mathbf{v}_2$ represents Galilean boost $u_2 = f(t, x - \epsilon t) + \epsilon$, $\mathbf{v}_3$ represents time translation $u_3 = f(t - \epsilon, x)$, and $\mathbf{v}_4$ represents space translation $u_4 = f(t, x - \epsilon)$.

## I.3. Reaction-Diffusion System

A reaction-diffusion system describes the behavior of chemical substances or biological patterns that undergo both reaction and diffusion. We consider a $\lambda$-$\omega$ reaction-diffusion system (Champion et al., 2019) governed by:

$$\begin{cases} u_t = (1 - (u^2 + v^2))u + \beta(u^2 + v^2)v + d_1(u_{xx} + u_{yy}), \\ v_t = -\beta(u^2 + v^2)u + (1 - (u^2 + v^2))v + d_2(v_{xx} + v_{yy}), \end{cases} \tag{86}$$

with $d_1 = 0.1$, $d_2 = 0.1$, and $\beta = 1$. The experimental setup is consistent with the Schrödinger equation.

*Figure 8.* Visualization result of symmetry discovery on reaction-diffusion system by LieNLSD. The first subplot shows the last 6 singular values. The other five subplots display the infinitesimal generators corresponding to the nearly-zero singular values after sparsification.

We present the visualization result of LieNLSD on reaction-diffusion model in Figure 8. Although LaLiGAN discovers the $SO(2)$ symmetry in the latent space of this system, it cannot explicitly provide the group action in the observation space. LieNLSD takes the independent variables into account and directly discovers the $SO(2)$ symmetry in the observation space. Specifically, LieNLSD obtains 5 nearly zero singular values, which indicates that the number of infinitesimal generators is 5. The corresponding explicit expressions are shown in Table 6. Among these, $\mathbf{v}_1$ represents rotation in the complex space, $\mathbf{v}_2$ represents space rotation, $\mathbf{v}_3$ represents time translation, and $\mathbf{v}_4, \mathbf{v}_5$ represent space translation.

