# OpenReview forum: "Explicit Discovery of Nonlinear Symmetries from Dynamic Data"
_ICML.cc/2025/Conference — ICML 2025 poster_

### Official Review · Reviewer_TKxK · 2025-03-11

**Overall Recommendation:** 3

**Summary:**

The paper proposes a method for detecting non-linear symmetries in data. In a nutshell, the authors solve this by first defining a base for the Lie algebra space (which is the space of infinitesimal generators), then solve the matrix equation of which of the linear combinations of these generators align with the "derivative of the data". The system the authors needs to solve remains linear because they are working at the level of the infinitesimal generators.

**Claims And Evidence:**

The authors claim that LieNLSD is, to the best of their knowledge, "... the first method capable of determining the number of infinitesimal generators with nonlinear terms and their explicit expressions."

I am not familiar enough with the literature to determine whether this is true or not.

**Essential References Not Discussed:**

See above

**Experimental Designs Or Analyses:**

The experiments are interesting, but I am not sure how standard they are in the field. See the Question section below.

**Methods And Evaluation Criteria:**

The experiments the authors propose are interesting, but I would be interested to understand if the method can work in other cases. See the Question section below.

**Other Comments Or Suggestions:**

See below.

**Other Strengths And Weaknesses:**

See below.

**Questions For Authors:**

My main question is related to the applicability domain of the methods. The experiments look very specifically designed to me. So, to make things concrete, could you apply your method to the following scenarios:
- SO(3) in R^3: I take data on S2 but presented in Cartesian coordinates (using the embedding S^2 \subset R^3), Can you discover the (local) generator of  SO(3) just from the data?
- if you take the rotated MNIST dataset, where each MNIST digit is randomly rotated (SO(2) symmetry). Can you discover SO(2) just from the data?

**Relation To Broader Scientific Literature:**

If my interpretation of the claim of the paper is correct, namely "a method for detecting non-linear symmetries in data", I see very interesting potential application of the method applied to generative modelling. There is a recent body of work that designs score-based generative modeling using Lie algebra (also these works and other could be included in the related work section):

- Bertolini et al, "Generative Modeling on Lie Groups via Euclidean Generalized Score Matching"
- Zhu et al, "Trivialized Momentum Facilitates Diffusion Generative Modeling on Lie Groups"
- Kim et al, "Maximum likelihood training of implicit nonlinear diffusion model"

**Theoretical Claims:**

I think the theoretical claims are well-defined and, to the degree I checked the math, correct.

I would be interested in seeing, however, a complete (toy?) example, where all the steps can be followed "by hand". This is very helpful when discussing implementation of theoretical constructs. I see that the authors have some parts of it in the appendix (SO(2)). Maybe for that case the authors can provide a whole setting and showing how their framework applies explicitly?

---

> ### Author Rebuttal · Authors · 2025-03-30
>
> Thank you for your careful reading and valuable feedback! Below we will address each of your concerns point by point.
>
> **Theoretical Claims**
>
> Taking the discovery of linear symmetries in the Heat equation $u_t=u_{xx}$ as an example. We specify the function library as $\Theta=[t,x,u]^T\in R^{3\times1}$. Then, the infinitesimal group action takes the form
> $$
> v=W\Theta\cdot\nabla=\Theta^T W_1\partial_t+\Theta^T W_2\partial_x+\Theta^T W_3\partial_u,
> $$
> where $W=[W_1,W_2,W_3]\in R^{3\times 3}$. Our ultimate goal is to solve for W.
>
> The algorithm first constructs $\Theta_n$. From Equations (7) and (8), we obtain $pr^{(2)}v=\phi^t\frac{\partial}{\partial u_t}+\phi^{xx}\frac{\partial}{\partial u_{xx}}+\dots$ (here, for simplicity, we omit irrelevant terms), where $\phi^t=-u_tD_t\Theta^T W_1-u_xD_t\Theta^T W_2+D_t\Theta^T W_3,\phi^{xx}=-(u_tD_{xx}\Theta^T+2u_{tx}D_x\Theta^T)W_1-(u_xD_{xx}\Theta^T+2u_{xx}D_x\Theta^T)W_2+D_{xx}\Theta^T W_3$. We rewrite it as
> $$
> pr^{(2)}v=
> \begin{bmatrix}
> -u_tD_t\Theta^T&-u_xD_t\Theta^T&D_t\Theta^T\\\\
> -(u_tD_{xx}\Theta^T+2u_{tx}D_x\Theta^T)&-(u_xD_{xx}\Theta^T+2u_{xx}D_x\Theta^T)&D_{xx}\Theta^T
> \end{bmatrix}
> vec(W)\cdot\nabla=\Theta_2vec(W)\cdot\nabla
> $$
> Therefore, we get $\Theta_2$ in Equation (9). Substituting $D_t\Theta=[1,0,u_t]^T,D_x\Theta=[0,1,u_x]^T,D_{xx}\Theta=[0,0,u_{xx}]^T,u_t=u_{xx}$, we have
> $$
> \Theta_2=
> \begin{bmatrix}
> -u_{xx}&0&-u_{xx}^2&-u_x&0&-u_xu_{xx}&1&0&u_{xx}\\\\
> 0&-2u_{tx}&-(u_{xx}^2+2u_{tx}u_x)&0&-2u_{xx}&-3u_xu_{xx}&0&0&u_{xx}
> \end{bmatrix}
> $$
> The PDE can be expressed as $F=u_{xx}-u_t=0$, so its Jacobian matrix with respect to $u_t,u_{xx}$ is $J_F=[-1,1]$. Substituting into Equation (12), we have $[u_{xx},-2u_{tx},-2u_{tx}u_x,u_x,-2u_{xx},-2u_xu_{xx},-1,0,0]vec(W)=0$.
>
> Comparing the coefficients, we obtain the basis of vec(W) as $[2,0,0,0,1,0,0,0,0]^T,[0,0,0,0,0,0,0,1,0]^T,[0,0,0,0,0,0,0,0,1]^T$. Substituting into $v=W\Theta\cdot\nabla$, we get the infinitesimal generators $v_1=2t\partial_t+x\partial_x,v_2=x\partial_u,v_3=u\partial_u$. These correspond to the linear generators $v_2,v_4,v_5$ shown in Table 6 (in Appendix I).
>
> **Relation To Broader Scientific Literature**
>
> Thank you for your addition! We will include the following content in the Related Work section.
>
> A series of recent works have made outstanding contributions to extending diffusion models from Euclidean space to Lie groups [1,2,3]. Since our method can directly discover the infinitesimal generators of Lie groups from data, we look forward to future integration with these works, eliminating the need for prior knowledge of the group to guide their processes.
>
> **Questions For Authors**
>
> Both of these scenarios involve discovering linear symmetries from static systems. By setting the prolongation order n=0 and specifying the function library $\Theta(x)=x$ as linear terms, LieNLSD can cover them (see Appendix F for details). The core issue is how to define F in Algorithm 1 and estimate $J_F$, which we will discuss in detail below.
>
> (1) For a two-dimensional manifold in $R^3$, its implicit equation can be written as $F(x)=0,x\in R^3$, but it may not have an explicit equation (e.g., S2). Therefore, we use the $k$-NN to estimate the Jacobian matrix $J_F$. Specifically, we fit a tangent plane $J_F(x-x_0)=0$ to the manifold using the k neighboring points around the given point $x_0$, and assign a Gaussian kernel weight $w_i=\exp(-\\|x_i-x_0\\|^2/2)$ to each point to reduce the influence of more distant points.
>
> In practice, we set the number of neighboring points to k=5. The experimental results are shown below. We successfully discover the generators of SO(3).
>
> Singular value: $[57.4,36.9,36.6,36.5,36.2,36.1,5.1,4.6,4.2]$
>
> The Lie algebra basis (W) corresponding to the 3 smallest singular values:
> $$
> \begin{bmatrix}
> 7.58e^{-4}&5.58e^{-4}&-0.706\\\\
> 1.10e^{-3}&9.63e^{-4}&8.15\times 10^{-4}\\\\
> 0.708&-1.07e^{-4}&-7.22e^{-4}
> \end{bmatrix},
> \begin{bmatrix}
> -1.28e^{-3}&0.706&-1.89e^{-5}\\\\
> -0.709&5.09e^{-4}&-1.71e^{-3}\\\\
> 5.31e^{-4}&2.08e^{-3}&3.47e^{-4}
> \end{bmatrix},
> \begin{bmatrix}
> -1.18e^{-3}&2.27e^{-4}&-5.77e^{-5}\\\\
> -3.56e^{-3}&1.17e^{-4}&0.704\\\\
> -9.67e^{-4}&-0.710&1.21e^{-3}
> \end{bmatrix}
> $$
> (2) In this case, the group transformation acts on the coordinates $x\in R^2$, so we need to compute the derivative of the classifier with respect to the coordinates $D_xF$. We train a CNN on MNIST, which takes grayscale images I(x) as input. Automatic differentiation yields $D_IF$. We then estimate  $D_xI$ using central differences on the image and apply the chain rule to obtain $D_xF=D_IF\cdot D_xI$. The experimental results are shown below, which demonstrate that we successfully discover the SO(2) symmetry.
>
> Singular value: $[48.8,46.8,41.2,22.0]$
>
> The Lie algebra basis (W) corresponding to the smallest singular value:
> $$
> \begin{bmatrix}
> -2.90e^{-3}&-0.680\\\\
> 0.733&-1.05e^{-2}
> \end{bmatrix}
> $$
> **Reference**
>
> [1] arxiv.org/abs/2502.02513
>
> [2] arxiv.org/abs/2405.16381
>
> [3] arxiv.org/abs/2205.13699

---

### Official Review · Reviewer_LaJV · 2025-03-14

**Overall Recommendation:** 3

**Summary:**

This paper introduces LieNLSD, a novel method for discovering nonlinear symmetries from trajectory data. It addresses the limitations of previous methods that primarily focus on linear symmetries. LieNLSD aims to determine the number of infinitesimal generators and their explicit expressions. The method involves specifying a function library for coefficients of the lie algebra, showing its prolongation formula's linearity with respect to the library coefficient matrix, and solving a system of linear equations derived from the infinitesimal criterion using SVD. The authors also apply LADMAP for sparsification of infinitesimal generators. The method demonstrates strong results on  various dynamical systems, and improves the accuracy of neural PDE solvers when using discovered symmetries for data augmentation.

## update after rebuttal
My concerns were addressed and with the new ablations, I raised my score.

**Claims And Evidence:**

Most of the claims made by the paper appear to be substantiated by empirical evidence, while theoretical results appear to be well explained. However, limitations of the method remain completely unexplored and it is not clear whether the results were cherry-picked.

**Essential References Not Discussed:**

n/a

**Experimental Designs Or Analyses:**

Apart from the issues listed above, the analysis appears to be sound.

**Methods And Evaluation Criteria:**

The problems considered in this paper appear to be relevant and good examples for the probem considered. However, the settings consdiered here appear to be quite limited. To be precise, while the method here appears to be quite good, SVD and library based methods for symbolic discovery have been known to be susceptible to a multituted of issues, and particualrly the following questions need to be addressed (at least through ablations):

* How sensitive is the method to noise in the observations?
* Currently, Table 3 presents the discovered bases, but not the true bases - making it unclear if they are accurate.
* What happens when there is a dictionary (library) misspecification?
* What is the effect of number of samples of the trajectory on the discoverability?

**Other Comments Or Suggestions:**

see below.

**Other Strengths And Weaknesses:**

see above.

**Questions For Authors:**

* Line 23 column 2 - Please provide a reference for why that is the case.
* Line 55 column 1 - both W and \Theta are introduced for the first time. It would be beneficialy for the paper to provide a short describption of the method at the beginnign of the paper, where these can be introduced.
* Line 107 - this makes it seem that it is always possible to represent a group element as an exponential, but the exponential map may not be surjective.
* Where does the notation for eqns 1 and 2 come from? This seems to be a bit different to Olver.
* Line 227 "can be solved *for*."
* Line 236 column 2 - the mapping f is meant to be F?
* Why is KdV not included in Table 3 if it is used for point symmetry augmentation?
* How are the networks F trained?
* Line 402 column 2 - you claim that LPSDA is only applicable to 1-dim cases, but that does not appear to be the case in the paper considered. Could you elaborate what you mean by this?

**Relation To Broader Scientific Literature:**

There appear to be a few recent missed references on the usage of lie symmetries derived through the algebra in the context of neural PDE solvers, beyond augmentation. I list these below:

https://arxiv.org/abs/2310.17053
https://www.mdpi.com/2673-9984/5/1/13
https://arxiv.org/pdf/2410.02698
https://arxiv.org/pdf/2212.04100v1
https://arxiv.org/abs/2206.09299
https://arxiv.org/pdf/2502.00373
https://arxiv.org/abs/2311.04293

**Theoretical Claims:**

The theoretical claims appear to intuitively be correct, but I have not looked at the exact proofs.

---

> ### Author Rebuttal · Authors · 2025-03-30
>
> Thank you for your careful reading and valuable feedback! Below we will address each of your concerns point by point.
>
> **Methods And Evaluation Criteria**
>
> (1) See point (1) in Methods And Evaluation Criteria section of Rebuttal to Reviewer 1vcx.
>
> (2) According to the procedure in Section 2.4 of the textbook [1], given the expression of the PDE, we can mathematically derive its true infinitesimal generators:
>
> Burgers: $\partial_x,\partial_t,\partial_u,x\partial_x+2t\partial_t,2t\partial_x-x\partial_u,4tx\partial_x+4t^2\partial_t-(x^2+2t)\partial_u,\alpha(x,t)e^{-u}\partial_u$, where $\alpha_t=\alpha_{xx}$
>
> Wave: $v_1=\partial_x,v_2=\partial_y,v_3=\partial_t,v_4=-y\partial_x+x\partial_y,v_5=t\partial_x+x\partial_t,v_6=t\partial_y+y\partial_t,v_7=x\partial_x+y\partial_y+t\partial_t,v_8=(x^2-y^2+t^2)\partial_x+2xy\partial_y+2xt\partial_t-xu\partial_u,$$v_9=2xy\partial_x+(y^2-x^2+t^2)\partial_y+2yt\partial_t-yu\partial_u,v_{10}=2xt\partial_x+2yt\partial_y+(x^2+y^2+t^2)\partial_t-tu\partial_u,v_{11}=u\partial_u,v_\alpha=\alpha(x,y,t)\partial_u$, where $\alpha_{tt}=\alpha_{xx}+\alpha_{yy}$
>
> Schrodinger:
> $\partial_x,\partial_y,\partial_t,-y\partial_x+x\partial_y,-v\partial_u+u\partial_v,-2t\partial_t-x\partial_x-y\partial_y+u\partial_u+v\partial_v$
>
> Then, it is easy to verify that the infinitesimal generators discovered by LieNLSD, as shown in Table 3, are all correct.
>
> (3) See point (2) in Methods And Evaluation Criteria section of Rebuttal to Reviewer pziR.
>
> (4) See point (2) in Methods And Evaluation Criteria section of Rebuttal to Reviewer 1vcx.
>
> **Relation To Broader Scientific Literature**
>
> Thank you for your addition! We will include the following content in Related Work section:
>
> Some recent works have introduced symmetry into PINNs [2,3,4,7,8,9] or performed data augmentation based on PDE symmetries [5,6], significantly improving the performance of neural PDE solvers. We expect to combine our data-driven symmetry discovery approach with these works in the future to explore methods for PDE solving without requiring prior knowledge of symmetries.
>
> **Questions For Authors**
>
> (1) True symmetries of the wave equation are provided in point (2) in Methods And Evaluation Criteria section of the Rebuttal (calculation process see [1]). As mentioned in Related Work section (Lines 128-130, Column 2), existing works on linear symmetry discovery use Lie algebra representations to characterize symmetries. Then, according to Eqn (2), they can only discover linear generators of the form $\phi(v)=d\rho_X(v)x\cdot\nabla$. For the wave equation, $v_1,v_2,v_3$ (translation symmetry) and $v_8,v_9,v_{10}$ (special conformal symmetry) do not conform to this form, meaning these symmetries cannot, in principle, be discovered by such methods.
>
> (2) We will revise the sentence to "Then the infinitesimal group action can be expressed in the form of $W\Theta$. Our ultimate goal ...".
>
> (3) We will revise the sentence to "... with a basis $\\{v_1,v_2,\dots,v_r\\}$, in the neighborhood of the identity element, we have ...".
>
> (4) In Chapter 1.4 of the textbook [1], Eqn (1.46) defines the infinitesimal group action as $\psi(v)=\frac{d}{d\epsilon}|_{\epsilon=0}\Psi(\exp(\epsilon v),x)$. However, $\psi(v)$ is usually written in the form $-y\partial_x+x\partial_y=(-y,x)\cdot\nabla$ rather than (-y,x). In Chapter 1.3 of the textbook, the paragraph immediately following Eqn (1.4) also mentions that readers can regard $\partial_x$ as "place holders" or a special "basis". To avoid confusion, we add the partial differential operator $\nabla$ to the infinitesimal group action, as shown in Eqns (1) and (2) of our paper.
>
> (5) We will make the revision. Thank you!
>
> (6) For a 1st-order dynamic system, the PDE can be written as $F(x,u^{(n)})=f(u')-u_t=0$ (Line 238, column 1). By treating u' as the input and $u_t$ as the output, we can train a neural network to approximate f, thereby obtaining F.
>
> (7) Due to page limitations, the experimental results of the KdV equation are provided in Table 6 of Appendix I rather than in the main text.
>
> (8) As mentioned in the response to Question 6, we indirectly obtain F by training a neural network to fit f. The training details are provided in Appendix H.
>
> (9) From methods, the original paper [6] only uses the 1D KdV equation as a worked example to discuss the implementation process of data augmentation. The trigonometric interpolation they employ becomes more complex in higher-dimensional cases. From experiments, they only test on 1D evolution equations as mentioned in Section 4.1 of their paper. Moreover, their code implementation is also limited to 1D cases.
>
> **Reference**
>
> [1] Olver, Peter J. Applications of Lie groups to differential equations.
>
> [2] arxiv.org/abs/2310.17053
>
> [3] www.mdpi.com/2673-9984/5/1/13
>
> [4] arxiv.org/pdf/2410.02698
>
> [5] arxiv.org/pdf/2212.04100v1
>
> [6] proceedings.mlr.press/v162/brandstetter22a.html
>
> [7] arxiv.org/abs/2206.09299
>
> [8] arxiv.org/pdf/2502.00373
>
> [9] arxiv.org/abs/2311.04293

---

> > ### Comment · Reviewer_LaJV · 2025-04-03
> >
> > I want to thank the reviewers for their detailed responses. With the new ablations, I am happy to raise my score.

---

> > > ### Author Response · Authors · 2025-04-05
> > >
> > > We are delighted that our response has addressed your concerns. We sincerely appreciate your valuable suggestions and further recognition of our work!

---

### Official Review · Reviewer_1vcx · 2025-03-14

**Overall Recommendation:** 3

**Summary:**

The paper introduces a novel data-driven PDE symmetry discovery. In contrast to previous arts based on end-to-end symmetry discovery (Ko et al., Yang et al.,), the proposed methods applies a post-hoc nullspace analysis on the learned PDE operator to discover the subspace of infinitesimal symmetry generators that correspond to the Lie algebra basis. This strategy offers significant benefits that was impossible before:
1. The dimension of the symmetry is not a predefined hyperparameter but a discovered quantity
2. The generator's explicit differential operator representation could be discovered.

Furthermore, while not highlighted by the authors, the training is much simpler and easier as the symmetry discovery is separated from the governing equation learning. This is a clear advantage over previous arts that leveraged unstable adversarial objective (Yang et al.) or several hyperparameter-sensitive regularizers (Ko et al.).

**Claims And Evidence:**

The paper's core claim is that the explicit representation of the PDE symmetry generators could be discovered with a post-hoc nullspace analysis on the Jacobians of the prolonged PDE operator as in Eq.13. This claim is theoretically and empirically well-supported.

One major concern is the potential confusion or overstatement regarding the term 'nonlinear symmetry.' While the symmetry operators are nonlinear with regard to the coordinate variables, the PDE and the symmetry operators are linear w.r.t. the field. The proposed method **cannot discover generators that act nonlinearly** on the field as LaLiGAN does. The authors should clarify this in Table 1 to avoid overstatement.

**Essential References Not Discussed:**

Relevant works are appropriately cited.

**Experimental Designs Or Analyses:**

This has already been discussed in *Methods And Evaluation Criteria*.

**Methods And Evaluation Criteria:**

The method has been tested on four scientific PDE. The method could discover the correct infinitesimal generator in their explicit form. The accuracy of the discovered subspace is measured with the Grassman distance, which is also an important contribution. The accuracy of the discovered generator subspace is significantly better than LieGAN, which is natural because the proposed method employs exact SVD-based subspace discovery instead of inaccurate feed-forward discovery.

However, it is unclear how the proposed approach would behave when the sample point is not enough or the measurement is noisy. As the proposed method relies on higher order derivatives, it is expected that the model is vulnerable to noise or low-resolution measurements. Since LieGAN does not rely on higher order derivatives, the authors should also compare the proposed method against LieGAN with **noisy and/or insufficient samples**.

**Other Comments Or Suggestions:**

As mentioned earlier, the statement regarding nonlinear symmetry needs further clarification, because both the PDE and the group action are linear.

**Other Strengths And Weaknesses:**

1. The proposed method requires predefined function libarary.
2. The proposed method cannot be used for arbitrarily high order prolongations.
3. Higher order derivatives may not be available or accurate in real-world scenarios.

**Questions For Authors:**

I am willing to raise my score once my two major concerns are addressed.
1. Clarification regarding the term *nonlinear symmetry.*
2. Validation under noisy or insufficient number of sample points.

**Relation To Broader Scientific Literature:**

As mentioned in the paper, the discovered symmetry could be used to augment the data for neural PDE solvers. It could also be used to prune the redundant generators that are discovered by approaches that require predefined number of generators like LieGAN.

**Theoretical Claims:**

The theoretical claims appear to be valid.

---

> ### Author Rebuttal · Authors · 2025-03-30
>
> Thank you for your careful reading and valuable feedback! Below we will address each of your concerns point by point.
>
> **Claims And Evidence**
>
> Note that in our paper, the symmetry is defined on $X\times U$ (see the "Symmetries of differential equations" paragraph in Section 2, lines 82-88, column 2). Therefore, the symmetries we find can be nonlinear with respect to both the coordinates $X$ and the field $U$.
>
> However, from an implementation perspective, the types of PDE symmetries discovered by LieNLSD (ours) and LaLiGAN differ (perhaps this is what you are concerned about?). Taking $X=R^2,U=R$ as an example (e.g. $u(x,y)$ represents a planar image), the symmetries found by LieNLSD act pointwise on $X\times U=R^3$. Such symmetries are commonly referred to in the literature as "Lie point symmetries." On the other hand, the symmetries found by LaLiGAN are defined over the entire discretized field. Specifically, if $u$ is a field on a $100\times100$ grid, then LaLiGAN's symmetries act on $R^{100\times100}$ (see the Reaction-diffusion dataset in the original paper of LaLiGAN for details). For PDEs, the setting of Lie point symmetries (which act pointwise on both coordinates and the field) is more common, as the search space is significantly reduced compared to symmetries defined over the entire discretized field, and the physical interpretation is more intuitive.
>
> We will add the above comparison of the two types of symmetries to the related work section. Thank you!
>
> **Methods And Evaluation Criteria**
>
> (1) Noisy samples
>
> We add multiplicative noise $u'=u(1+\epsilon)$ to the variables in the training dataset to evaluate the robustness of LieNLSD against noise, where $\epsilon\sim N(0,\sigma^2)$. The quantitative comparison results (Grassmann distance) between LieNLSD and LieGAN for different noise levels $\sigma$ are as follows. Although the performance of LieNLSD declines as the noise level increases, its accuracy remains higher than that of LieGAN, which shows robustness to noise.
>
> |Dataset|LieNLSD|LieGAN|
> |-|-|-|
> |Top tagging ($\sigma=0$)|$\mathbf{(9.20\pm1.83)\times10^{-2}}$|$(2.51\pm0.41)\times10^{-1}$|
> |Top tagging ($\sigma=0.05$)|$\mathbf{(3.75\pm0.40)\times10^{-1}}$|$2.27\pm0.01$|
> |Top tagging ($\sigma=0.1$)|$\mathbf{(8.61\pm0.55)\times10^{-1}}$|$2.39\pm0.02$|
> |Burgers eqn ($\sigma=0$)|$\mathbf{(1.26\pm0.20)\times10^{-2}}$|$1.58\pm0.05$|
> |Burgers eqn ($\sigma=0.05$)|$\mathbf{(1.54\pm0.25)\times10^{-1}}$|$1.34\pm0.45$|
> |Burgers eqn ($\sigma=0.1$)|$\mathbf{(3.30\pm0.73)\times10^{-1}}$|$1.66\pm0.02$|
> |Wave eqn ($\sigma=0$)|$\mathbf{(1.40\pm0.01)\times10^{-2}}$|$2.36\pm0.15$|
> |Wave eqn ($\sigma=0.05$)|$\mathbf{(3.80\pm0.26)\times10^{-2}}$|$2.24\pm0.05$|
> |Wave eqn ($\sigma=0.1$)|$\mathbf{(1.55\pm0.64)\times10^{-1}}$|$2.24\pm0.26$|
> |Schrodinger eqn ($\sigma=0$)|$\mathbf{(8.62\pm1.31)\times10^{-2}}$|$2.22\pm0.05$|
> |Schrodinger eqn ($\sigma=0.05$)|$\mathbf{1.89\pm0.20}$|$2.48\pm0.28$|
> |Schrodinger eqn ($\sigma=0.1$)|$\mathbf{2.14\pm0.03}$|$2.46\pm0.16$|
>
> (2) Insufficient samples
>
> To evaluate the error impact of estimating high-order derivatives using central differences on low-resolution grid points (insufficient samples), we downsample the discrete points in the training dataset with a sampling interval of $s$. For the low-resolution dataset, we employ higher-precision central differences to estimate the derivatives. For example, we replace the first-order derivative calculation $u_x=[u(x+h)-u(x-h)]/(2h)$ (with a truncation error of $O(h^2)$) with $u_x=[-u(x+2h)+8u(x+h)-8u(x-h)+u(x-2h)]/(12h)$ (with a truncation error of $O(h^4)$). The results of the ablation study (Grassmann distance) are presented below.
>
> |Dataset|LieNLSD|LieGAN|
> |-|-|-|
> |Burgers eqn (s=1)|$\mathbf{(1.26\pm0.20)\times10^{-2}}$|$1.58\pm0.05$|
> |Burgers eqn (s=3)|$\mathbf{(2.56\pm0.48)\times10^{-1}}$|$1.31\pm0.45$|
> |Burgers eqn (s=5)|$\mathbf{(6.84\pm0.86)\times10^{-1}}$|$1.70\pm0.07$|
> |Wave eqn (s=1)|$\mathbf{(1.40\pm0.01)\times10^{-2}}$|$2.36\pm0.15$|
> |Wave eqn (s=3)|$\mathbf{(1.26\pm0.01)\times10^{-2}}$|$2.39\pm0.05$|
> |Wave eqn (s=5)|$\mathbf{(1.95\pm0.02)\times10^{-2}}$|$2.37\pm0.12$|
> |Schrodinger eqn (s=1)|$\mathbf{(8.62\pm1.31)\times10^{-2}}$|$2.22\pm0.05$|
> |Schrodinger eqn (s=3)|$\mathbf{2.08\pm0.07}$|$2.55\pm0.19$|
> |Schrodinger eqn (s=5)|$\mathbf{2.18\pm0.01}$|$2.52\pm0.21$|
>
> As the resolution decreases, LieNLSD still performs better than LieGAN. In fact, works on PDE symmetry discovery can hardly bypass the limitation that numerical derivatives on low-resolution data tend to have large errors. For example, when calculating the validity score, Ko et al. [1] also needed to compute numerical derivatives for the transformed data to evaluate whether it satisfies the PDE (see Section 4.2 of the original paper [1] for details). However, higher-precision difference methods can be introduced to mitigate the impact of low resolution.
>
> **Reference**
>
> [1] Ko et al. "Learning Infinitesimal Generators of Continuous Symmetries from Data."

---

> > ### Comment · Reviewer_1vcx · 2025-04-09
> >
> > Dear authors, I appreciate your response and for conducting additional experiments with noisy observations. My concerns have been addressed, and I would like to keep my accept opinion.

---

> > > ### Author Response · Authors · 2025-04-09
> > >
> > > We sincerely appreciate your high-quality review comments and your further support for our work! We are glad that our response has addressed your concerns. In light of this, we were wondering if you might consider raising the score accordingly? Thank you once again!

---

### Official Review · Reviewer_pziR · 2025-03-16

**Overall Recommendation:** 4

**Summary:**

In this paper, the authors proposed LieNLSD, a method for explicitly discovering nonlinear Lie group symmetries that are not represented by $\mathrm{GL}(n)$, from data governed by a PDE $ F(x, u^{(n)}) = 0$. The proposed method parameterizes a nonlinear infinitesimal group action $\mathbf{v}$ acting on $(x, u)$ using a symbolic regression technique $\mathbf{v} = W \Theta(x, u) \cdot \nabla$, where $\Theta(x, u)$ is a predefined symbolic library and $W$ is a learnable matrix. The method then constructs the prolonged version of this group action, which satisfies $\mathrm{pr}^{(n)} \mathbf{v} = \Theta_n(x, u^{(n)})\mathrm{vec}(W) \cdot \nabla$. Utilizing the fact that the prolonged group action must satisfy $ \mathrm{pr}^{(n)} \mathbf{v} [F(x, u^{(n)})] = 0$ whenever the PDE $F(x, u^{(n)}) = 0$ exhibits symmetry under this group, the authors derive a natural learning criteria $J_F(x, u^{(n)}) \Theta_n (x, u^{(n)}) \mathrm{vec}(W) = 0$.

To find Jacobians $J_F$ from data pairs $(x[i], u[i])$ originating from an unknown PDE, the authors first construct derivatives $u^{(n)}[i]$ using the central difference method, then approximate the PDE through a neural network $f$, assuming a form of $u_t$ (or $u_{tt}$ or $0$) $= f(u’) \implies F(x, u^{(n)}) = f(u’) – u_t = 0$. Using the Jacobian approximated via automatic differentiation of the neural network $f$, the authors solve $J_F(x, u^{(n)}) \Theta_n (x, u^{(n)}) \mathrm{vec}(W) = C \mathrm{vec}(W) = 0$ through SVD of $C$ (which means that the method automatically identifies the Lie algebra dimension based on the number of zero singular values) , with additional sparsifications.

The authors benchmark their proposed approach with top quark tagging and PDE datasets, and showcase the use-case of the extracted symmetries to enhance the performance of neural PDE solvers.

**Claims And Evidence:**

The authors' claims regarding the advancements of their proposed method are clearly summarized in Table 1. The algorithm design is carefully structured to support the authors' objectives. Additionally, the claims are empirically validated through experimental results. For instance, the proposed method accurately identifies nonlinear symmetries from data across three different PDEs. Furthermore, its capability to automatically estimate the dimension of the Lie algebra is demonstrated using the top quark dataset, in contrast to LieGAN, which treats the Lie algebra dimension as a hyperparameter.

However, there are some concerns that need to be addressed to further support the authors' claims, which will be elaborated in the Method and Experimental Designs sections.

**Essential References Not Discussed:**

The authors provide a comprehensive list of references and discuss related work appropriately.

**Experimental Designs Or Analyses:**

The authors benchmark the proposed method using a top quark dataset and three PDEs, comparing it against the linear symmetry discovery model (LieGAN). Additionally, the nonlinear symmetry extracted by the proposed method is utilized to enhance the accuracy of the neural Fourier operator (FNO) by augmenting its training data with Lie point symmetries. The results demonstrate that the discovered symmetry can significantly improve the performance of FNO, achieving accuracy nearly comparable to augmentation with ground-truth symmetries. This finding is highly promising.

However, there is a concern that although the proposed method and benchmarks emphasize nonlinear symmetry, the authors do not provide comparisons with other nonlinear approaches, such as LaLiGAN and the method by Ko et al. The authors state that this omission is due to their choice of Grassmann distance as the evaluation metric, but this reasoning alone does not fully justify the absence of comparisons with these established nonlinear methods. If possible, I strongly recommend that the authors perform additional comparisons with these existing approaches using alternative, suitable metrics.

**Methods And Evaluation Criteria:**

This paper presents a rigorously justified method based on a solid mathematical formulation for nonlinear symmetry discovery. Leveraging Lie group theory, the proposed method is technically sound and principled. However, there are two potential concerns regarding the proposed method:

- The Jacobian $J_F$ is approximated by training a neural network surrogate to learn the PDE directly from observed data. Consequently, the accuracy of the identified symmetry depends on the performance of this neural surrogate. Therefore, the robustness of this approach is uncertain due to two primary sources of noise: (1) numerical derivative estimation based on the central difference scheme, and (2) approximation errors inherent in the neural network itself.

- The symbolic library $\Theta$ should be predefined based on prior knowledge of the problem. The automatic selection of the Lie algebra dimension and the explicit construction of the Lie algebra basis come at the cost of manually selecting the symbolic library. A library that is too limited may fail to capture accurate symmetries, while an excessively large library increases both the search space and computational complexity. It can be particularly problematic when there is no information about the governing equation at all.

Therefore, I believe the authors should discuss these aspects and conduct ablation studies to assess the robustness of the proposed method, particularly its sensitivity to noise and dependence on library selection.


For evaluation, the authors employ the Grassmann distance to measure similarity between the ground truth and the discovered Lie algebra subspace. While this metric is theoretically principled, its use inherently limits comparison with other nonlinear symmetry discovery methods that do not explicitly produce a subspace representation.

**Other Comments Or Suggestions:**

A small typo in line 199: $\mathrm{diag}[\Theta(x, u)^{T}, \dots, \Theta(x, u)^{T}] \in \mathbb{R}^{((p+q)\cdot r) \times ((p+q)\cdot r)}$.

**Other Strengths And Weaknesses:**

The paper is well-motivated, and the proposed method is technically sound. Furthermore, it is clearly written and easy to follow, particularly given the depth of the mathematical content. However, I believe the authors should address the concerns I raised in the Method and Experimental Design sections for further clarity.

**Questions For Authors:**

Please refer to the Method and Experimental Designs sections for major questions. Additionally, a minor question to consider:

Consider a real-world challenging scenario in which the governing PDE is entirely unknown, introducing uncertainties in both Jacobian estimation and library selection. Under such circumstances, is there an auxiliary metric or criterion available to evaluate the physical plausibility of the estimated symmetry?

**Relation To Broader Scientific Literature:**

This paper is highly relevant for identifying symmetries in PDEs, a fundamental task across various scientific fields.

**Theoretical Claims:**

This paper is not a theoretical study but primarily presents an algorithm for discovering nonlinear symmetry groups. The algorithm is constructed based on two theorems: Theorem 4.1 and Theorem 4.2. While Theorem 4.2 is drawn from a textbook, the proof of Theorem 4.1 is provided in the paper. I briefly reviewed the proof of Theorem 4.1 and found no significant flaws.

---

> ### Author Rebuttal · Authors · 2025-03-30
>
> Thank you for your careful reading and valuable feedback! Below we will address each of your concerns point by point.
>
> **Methods And Evaluation Criteria**
>
> (1) See the Methods And Evaluation Criteria section of the Rebuttal to Reviewer 1vcx.
>
> (2) We conduct ablation experiments using a limited $\Theta$ (including only linear and constant terms) and a large $\Theta$ (additionally including sine and exponential terms) to evaluate the robustness of LieNLSD with respect to the choice of function library. The quantitative comparison results (Grassmann distance) with LieGAN are presented below.
>
> |Dataset|LieNLSD (limited $\Theta$)|LieNLSD|LieNLSD (large $\Theta$)|LieGAN|
> |-|-|-|-|-|
> |Burgers' equation|$(1.30\pm0.23)\times10^{-2}$|$(1.26\pm0.20)\times10^{-2}$|$(1.06\pm0.14)\times10^{-2}$|$1.58\pm0.05$|
> |Wave equation|$(1.41\pm0.01)\times10^{-2}$|$(1.40\pm0.01)\times10^{-2}$|$(1.43\pm0.03)\times10^{-2}$|$2.36\pm0.15$|
> |Schrodinger equation|$(3.13\pm0.91)\times10^{-2}$|$(8.62\pm1.31)\times10^{-2}$|$(8.74\pm0.65)\times10^{-2}$|$2.22\pm0.05$|
> |Heat equation|$(3.08\pm1.29)\times10^{-3}$|$(7.39\pm1.04)\times10^{-4}$|$(1.31\pm0.51)\times10^{-3}$|$2.59\pm0.04$|
> |KdV equation|$(9.66\pm3.39)\times10^{-3}$|$(5.24\pm1.55)\times10^{-3}$|$(1.12\pm0.41)\times10^{-2}$|$1.56\pm0.00$|
> |Reaction-diffusion system|$(8.23\pm1.13)\times10^{-2}$|$(1.24\pm0.18)\times10^{-1}$|$(1.24\pm0.13)\times10^{-1}$|$1.11\pm0.11$|
>
> For large $\Theta$, although we misspecify several terms, LieNLSD can still obtain accurate results within a large search space. For limited $\Theta$, LieNLSD may indeed miss some generators, but for all generators whose terms are fully included in $\Theta$, it can still accurately discover them (as shown in the table above, LieNLSD with $\Theta$ containing only linear and constant terms can correctly identify all linear symmetries).
>
> **Experimental Designs Or Analyses**
>
> Although the Grassmann distance is not applicable to the method of implicit symmetry discovery, we additionally compare the long rollout test NMSE of FNO with LPSDA based on LieNLSD and FNO with LPSDA based on Ko et al. [1]. For Ko et al., while they cannot obtain explicit expressions for the infinitesimal generators, feeding a given point into their trained MLP still yields the specific values of the infinitesimal generators at that point, thereby guiding data augmentation. Note that for PDEs, the types of symmetries found by LaLiGAN and our method differ (details can be found in the Claims and Evidence section of the Rebuttal to Reviewer 1vcx), making it impossible to establish a unified metric. The experimental results are presented below. Ko et al. can improve the accuracy of FNO, but not as significantly as our method. In addition to quantitative advantages, Ko et al. require the explicit expression of the PDE to compute the validity score (see Section 4.2 of their original paper), whereas our LieNLSD does not rely on this prior knowledge.
>
> |Dataset|FNO+$\emptyset$|FNO+LieNLSD|FNO+GT|FNO+Ko et al.|
> |-|-|-|-|-|
> |Burgers' equation|$(2.33\pm1.07)\times10^{-4}$|$(1.80\pm0.56)\times10^{-4}$|$\mathbf{(1.75\pm0.37)\times10^{-4}}$|$(1.83\pm0.25)\times10^{-4}$|
> |Heat equation|$(1.07\pm0.10)\times10^{-1}$|$\mathbf{(5.99\pm0.04)\times10^{-2}}$|$(6.01\pm0.20)\times10^{-2}$|$(1.04\pm0.14)\times10^{-1}$|
> |KdV equation|$(1.74\pm0.10)\times10^{-1}$|$\mathbf{(1.42\pm0.16)\times10^{-1}}$|$(1.47\pm0.02)\times10^{-1}$|$(1.58\pm0.14)\times10^{-1}$|
>
> **Other Comments Or Suggestions**
>
> $\Theta(x,u)\in\mathbb{R}^{r\times 1}$, then $\Theta(x,u)^\top\in\mathbb{R}^{1\times r}$. By repeating it $(p+q)$ times and arranging them diagonally, we obtain $diag[\Theta(x,u)^\top,\dots,\Theta(x,u)^\top]\in \mathbb{R}^{(p+q)\times((p+q) \cdot r)}$. We have double-checked that it is not a typo, but thank you for your reminder!
>
> **Questions For Authors**
>
> As mentioned in the Experimental Designs Or Analyses section of the Rebuttal, the accuracy of FNO with LPSDA can serve as one metric. In fact, the accuracy of many recent symmetry-informed neural PDE solvers (detailed in the Relation To Broader Scientific Literature section of the Rebuttal to Reviewer LaJV) can also be used to measure the quality of symmetry. Essentially, these use the performance on downstream tasks after applying symmetry as a metric.
>
> Additionally, following the approach of LieGAN and LaLiGAN, a discriminator can be employed to quantify the distributional differences between the original and transformed data. These metrics do require some training overhead, but we believe this is nearly unavoidable in scenarios where the PDE is entirely unknown, as evaluation can only be conducted based on the dataset.
>
> **Reference**
>
> [1] Ko, Gyeonghoon, Hyunsu Kim, and Juho Lee. "Learning Infinitesimal Generators of Continuous Symmetries from Data." arXiv preprint arXiv:2410.21853 (2024).

---

> > ### Comment · Reviewer_pziR · 2025-04-04
> >
> > Thank you very much for the authors’ detailed responses and the additional experiments. The robustness tests, the ablation study on the basis set size, and the downstream performance comparison with Ko et al. notably enhance the practical significance of the paper. I am happy to raise my review score accordingly.

---

> > > ### Author Response · Authors · 2025-04-05
> > >
> > > We are delighted that our response has addressed your concerns. We sincerely appreciate your valuable suggestions and further recognition of our work!

---

### Decision · Program_Chairs · 2025-05-01

**Decision:**

Accept (poster)

**Comment:**

The paper proposes a method for detecting non-linear symmetries in data. The authors solve this by first defining a base for the Lie algebra space (which is the space of infinitesimal generators), then solve the matrix equation of which of the linear combinations of these generators align with the "derivative of the data". The system the authors needs to solve remains linear because they are working at the level of the infinitesimal generators. The discovered symmetry could be used to augment the data for neural PDE solvers. It could also be used to prune the redundant generators that are discovered by approaches that require predefined number of generators like LieGAN. The methods are demonstrated on important PDEs.